# BOOST POST-TRAINING QUANTIZATION VIA NULL SPACE OPTIMIZATION FOR LARGE LANGUAGE MODELS

## ABSTRACT

Existing post-training quantization methods for large language models (LLMs) offer remarkable success. However, the increasingly marginal performance gains suggest that existing quantization strategies are insufficient to support the development of more compressed models. To inspire new directions for future research, this paper introduces the concept of **null space** into LLMs quantization. We argue that the quantization error can be effectively alleviated by constraining the post-quantization weight perturbation to lie within the null space of input activations. To prove this fresh idea, we propose an intuitive example projection method on several PTQ baselines to validate whether the performance will be further improved. Specifically, we devise an efficient and accurate null space projection approximation tailored to the characteristics of LLMs, and then theoretically derive a closed-form solution for an equivalent vector of the obtained projection matrix to satisfy practical inference condition. When validating our method on several milestone PTQ baselines, further performance improvements can be noticed obviously, demonstrating the novel perspective of null space optimization for LLMs quantization is effective. We view this paper the first step to alleviate the quantization error based on the insights of null space, hoping it inspiring future researchers to design more advanced quantization methods. Codes are available at `https://anonymous.4open.science/r/q2n-2236`

## 1 INTRODUCTION

Large language models (LLMs) (Touvron et al., 2023; Achiam et al., 2023; Bai et al., 2023) have demonstrated remarkable performance across various tasks in recent years. However, billions of parameters in LLMs incurs significant storage and inference overheads. To address these issues, quantization (Li et al., 2021; Dettmers et al., 2022; Nagel et al., 2020), which reduces memory requirements and accelerates inference by converting high-precision values in LLMs into low-bit representations, receives tremendous attention. Existing quantization approaches can be divided into Quantization-aware Training (QAT) (Liu et al., 2023; Xu et al., 2024; Du et al., 2024) and Post-training Quantization (PTQ) (Xiao et al., 2023; Lin et al., 2024; Yuan et al., 2023) according to the pipeline. Between them, PTQ is more popular in LLM community because of its efficiency and resource-friendliness. Currently, PTQ techniques for LLMs can achieve lossless performance at 4-bit (Frantar et al., 2022; Shao et al., 2023; Zhao et al., 2024) and high-accuracy inference at 1.61-3 bits (Chee et al., 2023; Zhao et al., 2025b; Huang et al., 2024).

Despite the development of various advanced quantization strategies, they all suffer from a shared limitation. As known, all existing methods are uniformly motivated by the goal of minimizing quantization error. For example, the weight-only quantization error is formulated by $\|WX - W_qX\|_2^2$ (Zhao et al., 2025a). From a numerical perspective, regardless of how effective the quantization method is, the error is fundamentally unavoidable. Without additional constraints, such numerical error $(W - W_q)$ is bound to negatively affect the final output. Recognizing this limitation, we systematically consider the following question: ***Given the inevitability of numerical error, is there a way to alleviate its impact on the final quantization error?***

Aiming at this question, in this paper we propose a pioneering perspective that quantization error can be effectively alleviated through the theoretical properties of **Null Space** (Coleman & Pothen, 1986; 1987; Ravfogel et al., 2020). Specifically, for weight-only quantization, we aim to prove that as long as the weight perturbation $(W - W_q)$ lies in the null space of the input activations, the final quantization error will be alleviated effectively ($\|(W - W_q)X\|_2^2 \approx 0$). Here we select a straightforward method to prove this perspective: constructing an approximate projection for quantized model that maps the numerical error into the null space. If performance enhancements appear, this insight will be proved to be effective.

During validation, we face two main challenges. Typically, computing the null space projection relies on SVD decomposition (Wang et al., 2024), where the left singular vectors corresponding to zero singular values span the null space (Fang et al., 2024; Tang et al., 2025). However, the large matrix dimensions make SVD computation extremely expensive, and the singular values of activation matrices rarely exhibit exact zeros. To address these issues, we propose an efficient and accurate approximation method to get the null space projection $\Delta$ based on the Prefix-Suffix Sum Ratio of singular values. In addition, since there only exists $W_q$ during inference for a quantized model, simply applying $(W - W_q)\Delta X \approx 0$ is not practically meaningful, and storing $\Delta$ would incur additional memory overhead. Considering this, we reformulate the null space optimization as solving $W - \alpha W_q = (W - W_q)\Delta$ and derive a closed-form solution for the equivalent projection vector $\alpha$, where $\alpha$ can be easily absorbed into the scaling factors to avoid extra memory costs while achieving null space optimization.

We integrate our example approximation method with several milestone PTQ baselines to validate whether the performance will be further improved. As the results in Section 4, consistent enhancements occur on various LLMs and tasks, clarifying that alleviating quantization error based on null space makes great sense. **We view this paper as the first step towards alleviating quantization error based on the insights of null space. Rather than performance enhancement at present, our work introduces a fresh perspective and novel direction for future quantization development.**

## 2 RELATED WORKS

### 2.1 QUANTIZATION FOR LLMS

Quantization is one of the most widely studied model compression techniques (Frantar & Alistarh, 2023; Gou et al., 2021; Hu et al., 2022), offering both high performance and high compression ratios. According to the pipeline, existing approaches fall into two categories: Quantization-aware training (QAT) and Post-training quantization (PTQ). QAT (Liu et al., 2023; Wang et al., 2023; Ma et al., 2024) requires training from scratch and updating the weights, which can achieve higher performance generally but the increased training overheads significantly hinders its development.

In contrast, PTQ only need a small scale calibration set to get effective quantization parameters, so its efficiency and resource-friendliness have made it more popular in LLM community. GPTQ Frantar et al. (2022) leverages second-order information to dynamically update the remaining weights during quantization. AWQ (Lin et al., 2024) assesses the saliency of different weight channels based on the input activations and allocates appropriate scaling factors accordingly. QuIP (Chee et al., 2023) utilizes incoherence preprocessing to transform the weights, achieving high performance at 2-bit. Despite their success, all existing methods suffer from inevitable performance degradation caused by conventional quantization error formulation, demonstrating imperative extra optimization constraint.

### 2.2 NULL SPACE CONSTRAINT LEARNING

Null space is a classical concept in linear algebra which is extensively studied in mathematics (Frittelli et al., 1997), and recently it is increasingly applied in machine learning (Zhang et al., 2016). Adam-NSCL (Wang et al., 2021) introduces null space optimization into continual learning by forcing the network parameter update lying in the null space of the input feature to balance plasticity and stability. LoRA-Null (Tang et al., 2025) builds adapters initialized from the null space of the pretrained knowledge acitvation to encounter catastrophic forgetting in model finetuning. In knowledge editing, AlphaEdit (Fang et al., 2024) leverages this theoretical insights to balance knowledge-

update error and knowledge-preservation error. Inspired by these advances, for the first time we introduce null space theory into model quantization to alleviate the impact of quantization error.

## 3 METHOD

In this section, we first introduce the concept of the null space and how it helps alleviate the quantization error. Subsequently, we introduce our proposed null space projection method to validate this perspective. Specifically, we claim that the conventional SVD-based method to get null space projection is not practical for LLMs, and then propose an efficient and accurate example null space approximation method accordingly. Finally, to satisfy practical inference, we redefine the objective of null space optimization and derive a closed-form solution for the equivalent projection vector $\alpha$.

### 3.1 NULL SPACE OPTIMIZATION ALLEVIATES QUANTIZATION ERROR

Model quantization aims to convert high-precision values into corresponding low-bit formats to reduce inference overheads. For LLMs, the weight-only quantization function can be elaborated as:

$$W_q = s(\text{clamp}(\lfloor \frac{W}{s} \rceil + z, 0, 2^b - 1) - z), \tag{1}$$

where $W \in \mathbb{R}^{n \times m}$ and $W_q \in \mathbb{R}^{n \times m}$ indicate full-precision and quantized weights respectively. $\lfloor \cdot \rceil$ denotes round-to-nearest operator. $s$ is the scaling factor and $z$ is the zero-point. Then, the objective of all existing weight-only quantization approaches is to minimize the squared error of full-precision and quantized outputs, as formulated by:

$$\underset{W_q}{\arg\min} \|WX - W_qX\|_2^2. \tag{2}$$

However, using this objective alone leads to a common issue across all existing methods: regardless of how advanced the quantization algorithm is, a numerical discrepancy from the full-precision weights will always exist. Without additional constraints, the numerical error $(W - W_q)$ will inevitably degrade the final performance, which will be more serious at low-bit (2-3 bits) scenario. This observation motivates us to ask the question: Considering that the numerical error is unavoidable, can we instead alleviate its impact by some strategy?

Null space, a classical theory in linear algebra, enters our view, which is defined as follows: *Given two matrices A and B, A lies in the null space of B if and only if $AB = 0$.* Based on it, we reconsider the question above and present the following lemma:

**Lemma 3.1** *If the numerical error $(W - W_q)$ lies in the null space of input activation $X$, the quantization error formulation will be changed into:*

$$\|WX - W_qX\|_2^2 = \|(W - W_q)X\|_2^2 \approx 0. \tag{3}$$

Lemma 3.1 implies that **as long as the weight perturbation induced by quantization lies within the null space of the input activations, the quantization error will be significantly alleviated**. Therefore, in this paper we aim to prove this perspective effective by devising an effective and accurate projection $\Delta$ for the numerical error $(W - W_q)$, achieving post-quantization optimization.

### 3.2 EFFICIENT AND ACCURATE APPROXIMATION FOR NULL SPACE PROJECTION

In previous section, we establish that if the weight perturbation after quantization lies into the null space of the input activations, the quantization error can be effectively mitigated. Next, we aim to validate it by providing a post-quantization null space projection.

Following existing methods (Wang et al., 2021; Fang et al., 2024), the layer-wise null space of input activation $X$ can be modeled as that of their uncentered covariance matrix $XX^T$ to guarantee stability, whose null space is equal to that of $X$ (please refer to Appendix for detailed proof). Subsequently, the first step of the conventional method for conducting null space projection is to apply SVD decomposition to $XX^T$:

$$U, \Sigma, V = \textbf{SVD}(XX^T), \tag{4}$$

where $U/V$ and $\Sigma$ denote the left/right singular vector and the diagonal matrix with $r$ singular values ($r = \text{rank}(XX^T)$), respectively. Then, extract the column vectors in $U$ corresponding to zero singular values to conduct a submatrix $U_1$ (the remaining submatrix is $U_2$). With $U_1$, we can get the null space projection operator $\Delta = U_1 U_1^T$ (refer to Appendix for detailed proof), which satisfies:

$$\|(W - W_q)\Delta X\|_2^2 \approx 0. \tag{5}$$

Although this traditional approach achieves great success in prior scenarios, we identify several limitations when applying it to LLMs: (a) **The dimension of activation matrices in LLMs is much higher, making their SVD decomposition prohibitively slow**; (b) **the singular values of activation matrices in LLMs rarely reach exact zeros, hindering the derivation of null space projection**. To successfully adapt to LLMs quantization, we introduce the following improvements over the conventional null space projection derivation.

**Efficient Eigenvalue Decomposition**   Computing the null space projection only requires the singular values and left singular vectors. For large-scale real symmetric matrices $XX^T$, the eigenvectors are identical to the left singular vectors, and the singular values correspond to the absolute values of the eigenvalues. As a result, Eq. 4 can be reformulated as: $XX^T = U\Sigma U^T = Q\lambda Q^T$, where $Q$ and $\lambda$ are the eigenvectors and eigenvalues, respectively.

Although the two are mathematically related, in practice, computing SVD requires multiple Householder transformations and leverages implicit QR iterations to approximate the singular values, which result in a more complex computation pipeline. In contrast, eigenvalue decomposition allows for faster eigenvalue computation via the QR algorithm (Watkins, 1982), and efficient recovery of eigenvectors through inverse iteration. Moreover, PyTorch's backend acceleration framework (Paszke, 2019) employs a divide-and-conquer strategy specifically optimized for eigenvalue decomposition of real symmetric matrices to achieve higher computational efficiency. Therefore, we suggest estimating the null space for LLMs by eigenvalue decomposition via QR-based iteration instead of conventional SVD, which significantly improves decomposition efficiency. In Table 3 we report the improvements in decomposition efficiency and the corresponding accuracy comparison, demonstrating its superiority.

**Accurate Null Space Approximation**   It is hard to guarantee that there exists exact zero singular values in the activation matrices of LLMs. Moreover, we observe that the rank returned by PyTorch's built-in matrix rank estimation which implicitly ignores small singular values differs significantly from the fact, as shown in Figure 1, impeding the calculation of null space projection. To address this issue, in this part we introduce how to get the null space projection in the absence of exact zero singular values.

According to Principal Component Analysis, we can consider $U_2$ as the principal components ($XX^T = U_2\lambda U_2^T$). Because $U_1$ contains the vectors corresponding to all the smallest singular values in $\lambda$, the null space can be approximated by adaptively selecting the range of $U_1$. Motivated by it, we propose a novel accurate approximation method to get the null space projection based on the Prefix-Suffix Sum Ratio of singular values. Specifically, we observe that the first singular value in LLMs activations is much larger than the sum of the others, so we remove the first value to eliminate its impact. Subsequently, we select a threshold $t$, and then identify an index $k$ such that the ratio between the sum of eigenvalues after $k$ and the sum before $k$ does not exceed $t$, which can be elaborated as:

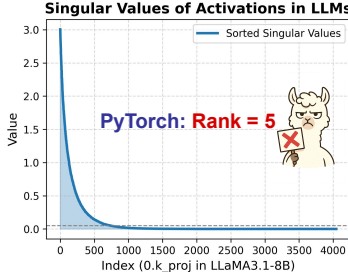

Figure 1: Singular values of Activations in LLMs.

$$R = \frac{\sum_{i=k+1}^{m} \lambda_i}{\sum_{i=1}^{k} \lambda_i} \leq t. \tag{6}$$

The ratio threshold $t$ is empirically set to 0.1. After identifying the index $k$ that satisfies the threshold condition, we redefine the range of $U_1$ accordingly to estimate the null space projection $\Delta$ based on $U_1 U_1^T$. With this projection, we can successfully map the quantization numerical error $(W - W_q)$ into the null space of $X$, which satisfies $\|(W - W_q)\Delta X\|_2^2 \approx 0$.

It is worth noting that (Wang et al., 2021) estimates the null space by selecting singular values that are 50 times larger than the smallest one. While effective for small-scale image features in CNNs,

their approach fails to provide reliable null space estimation for LLMs. Detailed comparison among PyTorch, Adam-NSCL and ours are provided in Table 4, demonstrating our method consistently leads to better performance.

### 3.3 CLOSED-FORM SOLUTION FOR THE EQUIVALENT VECTOR OF NULL SPACE PROJECTION

Although we get the null space projection efficiently and accurately, **there are only quantized weights $W_q$ and no full-precision weights $W$ for a real-quantized LLM during inference**, making Eq. 5 no longer practically meaningful. Moreover, storing $\Delta$, a large scale projection matrix, will incur additional memory overhead. To eliminate this challenge, in this subsection we focus on exploring a memory-free alternative of Eq. 5 to achieve the null space optimization directly on $W_q$.

To avoid additional memory overheads, the alternative must be integrated into existing components. As known, a real-quantized model involves not only low-bit weights but also full-precision channel-wise scaling factors. Therefore, we define **an equivalent projection vector** $\alpha$ for $W_q$, which fully avoids any extra memory costs by applying Hadamard product with the scaling factors.

Critically, to ensure the projection vector $\alpha$ applied to $W_q$ achieves the same effect as the original projection matrix $\Delta$ applied to the quantization numerical error $(W - W_q)$, we define the objective function as below:

$$\alpha^* = \underset{\alpha}{\arg\min} \|(W - W_q) \times \Delta - (W - \alpha W_q)\|_2^2. \tag{7}$$

It is important to note that $W_q$ has already undergone optimization by the quantization process, such as GPTQ. Directly imposing modification on $W_q$ through $\alpha$ may disrupt these prior optimizations. To address this, we augment Eq. 7 with a regularization term that ensures the result of null space optimization ($\alpha W_q$) remains close to the original fake-quantized weights ($W_q$). Concretely, we constrain the projection vector $\alpha$ to stay close to the unit vector. With this regularization, Eq. 7 is reformulated as:

$$\alpha^* = \underset{\alpha}{\arg\min}(\|(W - W_q) \times \Delta - (W - \alpha W_q)\|_2^2 + \lambda\|\alpha - 1\|_2^2), \tag{8}$$

where $\lambda$ is the regularization coefficient which is set to 0.2 empirically. The most straightforward approach for solving $\alpha^*$ is backpropagation. However, BP significantly increases the algorithm's complexity and hinders scalability. Considering this problem, we instead derive a closed-form solution by reformulating the objective function as a least squares problem. Since the quadratic term is strictly positive definite, the objective is strongly convex and admits a unique global optimum. By setting the gradient of the objective to zero, we obtain the closed-form solution for the optimal equivalent projection vector:

$$\alpha_i^* = \frac{\langle W_q^i, H^i \rangle + \lambda}{\langle W_q^i, W_q^i \rangle + \lambda}, \tag{9}$$

where $i$ is the i-th dimension and $H = W - (W - W_q) \times \Delta$. The detailed derivation is available in Appendix. Directly applying $\alpha^*$ to $W_q$ makes the null space optimization for quantized model practically meaningful, because it has a similar effect with projecting the quantization error using $\Delta$, while completely avoiding additional memory overheads. In Figure 3, we compare the performance of closed-form solution with the projection vector optimized by BP to prove our superiority.

The pipeline of our null space optimization for LLMs PTQ is summarized in Algorithm 1, which is named as Q2N, short for Quantize-to-Nullspace, to highlight the novel perspective compared to all previous works. In the following section, we leverage our Q2N to validate the effectiveness of alleviating quantization error via null space optimization.

## 4 EXPERIMENTS

In this section, we conduct experiments to address the following questions to prove the effectiveness of our null space optimization strategy for LLMs quantization:

• **RQ1:** Can null space optimization consistently improve the performance of a quantized LLM on both language generation and downstream reasoning tasks? Will calibration sets and model structure influence the effectiveness of null space optimization?

---

**Algorithm 1** Overall algorithm of our example null space optimization method for LLMs PTQ.

---

**Require:** full-precision weights $W$; input activation $X$; quantization method $Q(\cdot)$;
**Ensure:** quantized weights optimized by null space projection;
 1: *# Get the original quantized weights.*
 2: $W_q = Q(W)$.
 3: *# Compute the eigenvalue decomposition of of $XX^T$ efficiently.*
 4: $U, \lambda, U^T = \text{Eigen}(XX^T)$.
 5: *# Compute the Prefix-Suffix Sum Ratio of singular values based on threshold $t$ to get the index $k$.*
 6: $R = \frac{\sum_{i=k+1}^{m} \lambda_i}{\sum_{i=1}^{k} \lambda_i} \le t$.
 7: *# Get $U_1$ that corresponds to the last $k$ singular values.*
 8: $U_1 = U[:, k:]$.
 9: *# Compute null space projection $\Delta$.*
10: $\Delta = U_1 U_1^T$.
11: *# Compute the equivalent projection vector $\alpha$ for $W_q$.*
12: $H = W - (W - W_q) \times \Delta$.
13: $\alpha^* = \frac{\langle W_q^i, H^i \rangle + \lambda}{\langle W_q^i, W_q^i \rangle + \lambda}$.
14: **return** the final quantized weights optimized by null space projection $\alpha^* W_q$.

---

• **RQ2:** Will quantization strategy influence the effectiveness of null space optimization? Beyond weight-only quantization, is null space optimization also applicable to weight-activation schemes?

• **RQ3:** When proving null space optimization is effective for quantization, how much speedup does the efficient eigenvalue decomposition method offer compared to conventional SVD-based approaches, and does it affect the final performance?

• **RQ4:** Is our accurate null space approximation method more effective than Adam-NSCL and PyTorch's built-in matrix rank estimation function?

• **RQ5:** How does our closed-form solution for the null space projection vector compare with the more intuitive BP-based approach?

### 4.1 EXPERIMENTAL SETUP

**Models and Baseline Methods.** Our experiments are primarily conducted on LLaMA3 (8B, 70B), LLaMA3.1 (8B, 70B) and LLaMA3.3 (70B) (Grattafiori et al., 2024), as they are currently the most popular and widely applied open-sourced LLMs. In addition, three prominent star LLMs (DeepSeekMoE-16B (Dai et al., 2024), Qwen2.5-32B (Yang et al., 2024) and Qwen3-32B (Team, 2025)) are also included for evaluation. We integrate our Q2N with GPTQ (Frantar et al., 2022), the broadest practically deployed LLMs PTQ method, to valid different performance aspects. To ensure generality, we also select four baseline methods (QuIP (Chee et al., 2023), PB-LLM (Shang et al., 2023), LeanQuant (Zhang & Shrivastava, 2024) and QuaRot (Ashkboos et al., 2024)) which represent diverse quantization strategies according to (Zhao et al., 2025a).

**Implementation Details.** According to Section 3, quantization error becomes more pronounced at lower bit, so our null space optimization focus on 2(g128)/3-bit scenario. For accurate null space approximation and the closed-form solution of the projection vector, the threshold $t$ is set to $0.1$ and the regularization coefficient $\lambda$ is set to $0.2$ (optimal in most cases, please refer to Appendix for detailed). Our calibration set consists of 128 random 2048 token-segments from WikiText2 (Merity et al., 2016) and C4 (Raffel et al., 2020). All procedures are deployed on 1 A800-80G GPU.

**Evaluation Metrics.** We evaluate language generation capability (perplexity $\downarrow$) and downstream reasoning capability (zero-shot accuracy $\uparrow$) for the optimized quantized LLMs. For language generation tasks, the test data comes from WikiText2, PTB (Marcus et al., 1994) and C4. Downstream reasoning tasks includes ARC (Clark et al., 2018), HellaSwag (Zellers et al., 2019), Race (Lai et al., 2017), MMLU (Hendrycks et al., 2020), PIQA (Bisk et al., 2020) and WinoGrande (Sakaguchi et al., 2021), using the open-sourced toolkit lm-evaluation-harness (Gao et al., 2024) to evaluate.

Table 1: Evaluation results of different structural LLMs quantized by GPTQ (with Q2N or not) on language generation and downstream reasoning tasks (3-bit for 8B, 2-bit for the others).

| Model | Calib | Q2N | Language Generation (↓) | | | Downstream Reasoning (%, ↑) | | | | | | |
|---|---|---|---|---|---|---|---|---|---|---|---|---|
| | | | Wiki | PTB | C4 | ARC-c | ARC-e | HellaS | RACE | MMLU | PIQA | WinoG |
| LLaMA3-8B | Wiki | × | 18.74 | 35.87 | 35.74 | 26.02 | 37.46 | 52.71 | 30.72 | 24.28 | 59.58 | 59.19 |
| | | ✓ | **13.85** | **33.09** | **27.18** | **29.10** | **44.19** | **55.82** | **33.40** | **26.76** | **63.71** | **62.04** |
| | C4 | × | 19.83 | 33.96 | 23.08 | 24.66 | 40.03 | 48.72 | **35.12** | 29.93 | 60.01 | 62.04 |
| | | ✓ | **17.55** | **26.67** | **20.20** | **28.92** | **45.20** | **62.11** | 34.26 | **32.09** | **63.76** | **62.27** |
| LLaMA3.1-8B | Wiki | × | 15.67 | 36.44 | 26.32 | 31.91 | 50.25 | 51.25 | 34.26 | 24.19 | 66.00 | 61.96 |
| | | ✓ | **12.31** | **28.61** | **23.16** | **32.42** | **54.17** | **56.66** | **37.22** | **24.84** | **69.37** | **62.43** |
| | C4 | × | 24.55 | 36.72 | 26.79 | 31.23 | 49.71 | 58.70 | 34.07 | 32.25 | 66.27 | 60.06 |
| | | ✓ | **17.79** | **25.77** | **20.01** | **34.30** | **57.11** | **62.40** | **36.84** | **33.26** | **69.75** | **60.30** |
| LLaMA3-70B | Wiki | × | 16.40 | 29.21 | 32.00 | 26.19 | 37.58 | 49.24 | 31.67 | **25.98** | 61.70 | 56.51 |
| | | ✓ | **14.84** | **28.80** | **27.62** | **26.52** | **39.02** | **52.23** | **32.76** | 23.32 | **63.00** | **58.56** |
| | C4 | × | 22.39 | 30.17 | 26.89 | 25.26 | 36.41 | 50.10 | 32.15 | 25.72 | 60.01 | 56.91 |
| | | ✓ | **19.92** | **28.80** | **23.72** | **25.70** | **39.29** | **54.17** | **33.68** | **26.41** | **62.50** | **59.98** |
| LLaMA3.1-70B | Wiki | × | 14.36 | 26.53 | 27.41 | 28.24 | 43.27 | 53.04 | 31.87 | 25.71 | 63.44 | 57.30 |
| | | ✓ | **13.09** | **24.93** | **25.36** | **29.78** | **46.17** | **54.13** | **32.63** | **27.00** | **64.60** | **58.46** |
| | C4 | × | 18.97 | 27.61 | 21.97 | 29.95 | 45.08 | 57.53 | 33.40 | 25.28 | 67.25 | 58.33 |
| | | ✓ | **17.42** | **24.30** | **20.64** | **30.97** | **45.66** | **60.32** | **34.83** | **28.03** | **69.21** | **60.14** |
| LLaMA3.3-70B | Wiki | × | 15.27 | 28.33 | 31.18 | 31.31 | 44.87 | 57.66 | 36.84 | 27.94 | 66.27 | 59.59 |
| | | ✓ | **13.92** | **26.64** | **26.52** | **32.08** | **46.25** | **59.51** | **37.61** | **28.81** | **66.83** | **60.30** |
| | C4 | × | 18.34 | 25.04 | 21.67 | 31.91 | 46.63 | 59.49 | 36.65 | 34.56 | 68.44 | **61.33** |
| | | ✓ | **17.58** | **24.90** | **20.95** | **32.00** | **48.78** | **62.16** | **37.51** | **36.43** | **68.88** | **61.33** |
| DeepSeek-16B | Wiki | × | 27.36 | - | 107.81 | 24.40 | 31.94 | 33.97 | 25.26 | 23.83 | 55.60 | 50.36 |
| | | ✓ | **22.07** | - | **79.55** | **26.28** | **32.87** | **37.38** | **26.41** | **24.99** | **61.26** | **52.25** |
| | C4 | × | 48.45 | - | 48.34 | 23.81 | 35.40 | 38.93 | 28.90 | 23.49 | 60.34 | 50.59 |
| | | ✓ | **45.17** | - | **35.23** | **26.96** | **40.32** | **41.77** | **29.85** | **24.34** | **62.08** | **51.85** |
| Qwen2.5-32B | Wiki | × | 13.82 | 28.58 | 25.14 | 30.46 | 41.67 | 51.17 | 32.82 | 26.78 | 62.24 | 50.28 |
| | | ✓ | **12.29** | **24.68** | **22.01** | **31.80** | **46.30** | **54.80** | **36.65** | **27.36** | **63.22** | **50.91** |
| | C4 | × | 23.23 | 26.33 | 18.29 | 28.24 | 43.73 | 55.91 | 33.11 | 28.77 | 63.71 | 52.09 |
| | | ✓ | **15.83** | **23.51** | **17.83** | **31.91** | **44.99** | **60.82** | **33.88** | **30.77** | **67.90** | **52.70** |
| Qwen3-32B | Wiki | × | 22.89 | 56.08 | 35.45 | 27.82 | 33.92 | 43.01 | 28.71 | 24.40 | 53.54 | 52.72 |
| | | ✓ | **18.50** | **43.85** | **28.62** | **29.78** | **36.66** | **47.78** | **29.76** | **24.73** | **58.38** | **53.54** |
| | C4 | × | 37.10 | 54.97 | 26.59 | 25.43 | 32.45 | 47.85 | 30.21 | 24.88 | 57.50 | 51.46 |
| | | ✓ | **26.98** | **42.97** | **22.72** | **28.50** | **35.73** | **53.09** | **31.96** | **25.24** | **59.85** | **53.59** |

## 4.2 Performance on Language Generation and Downstream Reasoning (RQ1)

Language generation represents the most fundamental capability of LLMs, and accuracy on downstream reasoning tasks reflects their capacity for logical inference. To verify whether null space optimization can consistently improve the performance, we combine our Q2N with GPTQ, the most popular LLM quantization algorithm currently, to quantize several SOTA star open-sourced LLMs for evaluation. Specifically, LLaMA3/3.1-8B are quantized to 3-bit while the others are 2-bit with groupsize 128. From Table 1, we can observe that, regardless of the model, calibration set, or evaluation metric, incorporating Q2N consistently improves the performance. For example, LLaMA3-8B only achieves 48.72% on HellaSwag initially, while increasing to 62.11% after null space optimization. *Therefore, we can answer RQ1 that null space optimization consistently improve the performance on language generation and downstream reasoning task, and both calibration sets and model structure will not influence its effectiveness.*

## 4.3 Performance Enhancements on Different Strategies (RQ2)

To valid whether quantization strategy has negative impacts, followed by (Zhao et al., 2025a) we select three SOTA baseline methods from different PTQ strategies to combine with our Q2N: QuIP (Chee et al., 2023) represents rotation-based strategy, PB-LLM (Shang et al., 2023) typifies the combination of salience-based and compensation-based strategy, and LeanQuant (Zhang & Shrivastava, 2024) exemplifies the combination of optimization and compensation-based strategy. As indicated by Figure 2, all three baselines experience notable performance improvements. For example, QuIP achieves a 7.3 reduction in perplexity on WikiText2 at 2-bit, LeanQuant improves inference accuracy on HellaSwag by 10.7%.

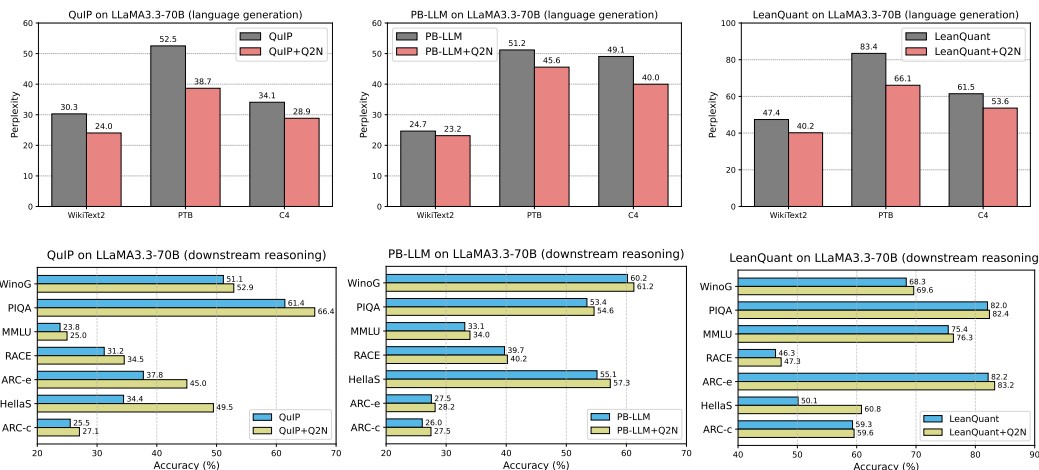

Figure 2: Performance enhancements when incorporating Q2N with the baselines of different strategies on LLaMA3.3-70B (2-bit, C4 Calib). Detailed results please refer to Appendix.

We also extend the scope to weight-activation (WA) PTQ scenario. Specifically, we select QuaRot (Ashkboos et al., 2024), the state-of-the-art WA method, to perform W4A4 quantization on LLaMA families, and then use Q2N to optimize the quantized weights after its original process. As demonstrated in Table 2, our Q2N further pushes the performance limits at each metric. *Therefore, we can answer RQ2 that null space optimization enables performance improvements on diverse quantization strategies, including the SOTA WA method.*

Table 2: Performance enhancement on QuaRot (W4A4, C4 Calib).

| LLaMA | Methods | Wiki | PTB | C4 |
|---|---|---|---|---|
| 3-8B | QuaRot | 8.57 | 13.60 | 11.95 |
|  | **+Q2N** | **8.51** | **13.41** | **11.88** |
| 3.1-70B | QuaRot | 6.48 | 11.67 | 10.24 |
|  | **+Q2N** | **6.39** | **11.55** | **10.19** |
| 3.3-70B | QuaRot | 7.39 | 13.48 | 11.54 |
|  | **+Q2N** | **7.29** | **13.08** | **11.39** |

## 4.4 EFFICIENT EIGENVALUE DECOMPOSITION *vs* SVD DECOMPOSITION (RQ3)

We decide to prove the effectiveness of null space optimization through a post-quantization projection. In Section 3.2, we establish that the large matrix sizes in LLMs makes the conventional SVD-based null space projection method extremely slow. In this part, we empirically demonstrate the speed and performance differences between SVD and our Efficient eigenvalue decomposition. Specifically, we replace our ef-

Table 3: Runtime and performance comparison between conventional SVD and our efficient decomposition method on LLaMA3.1-8B (3-bit, C4 Calib).

| Methods | Runtime (s) | | | | | | | Perplexity (↓) | | Avg.Acc (↑) |
|---|---|---|---|---|---|---|---|---|---|---|
| | Q | K | V | O | Up | Gate | Down | Wiki | C4 | |
| SVD | 4.14 | 4.15 | 4.14 | 3.16 | 3.48 | 3.54 | 142.84 | 23.24 | 23.36 | 43.95% |
| Q2N | 0.15 | 0.17 | 0.15 | 0.15 | 0.16 | 0.15 | 3.35 | 17.79 | 20.01 | 50.57% |

ficient decomposition component in Q2N with SVD and combine it with GPTQ to quantize LLaMA3.1-8B (3-bit, C4 calibration), and then record the per-layer runtime of both methods as well as their performance. As presented in Table 3, our efficient decomposition method achieves substantial runtime reductions across all linear layers compared to SVD, with speedups ranging from 21.07 (self_attn.o_proj) to 42.64 (mlp.down_proj). At the same time, it also outperforms SVD in terms of both perplexity and average accuracy.

## 4.5 ACCURATE NULL SPACE APPROXIMATION *vs* ADAM-NSCL & PYTORCH (RQ4)

In Section 3.2, we introduce an accurate null space approximation method based on the Prefix-Suffix Sum Ratio of singular values to address the challenge that the activations in LLMs rarely contain exact zero singular values. Previously, Adam-NSCL (Wang et al., 2021) computes the null space in continual learning for CNN-based image classification. In addition, PyTorch also provides a built-in matrix rank estimation function `Torch.linalg.matrix_rank()`, which implicitly ignores small singular values. In Table 4, we compare the three approaches on LLaMA3.1-8B (3-bit, C4

Table 4: Performance comparison among naive GPTQ, PyTorch, Adam-NSCL and our accurate null space approximation method on LLaMA3.1-8B (3-bit, C4 calibration).

| Methods | Language Generation (↓) | | | Downstream Reasoning (%, ↑) | | | | | | |
|---|---|---|---|---|---|---|---|---|---|---|
| | Wiki | PTB | C4 | ARC-c | ARC-e | HellaS | RACE | MMLU | PIQA | WinoG |
| GPTQ | 24.55 | 36.72 | 26.79 | 31.23 | 49.71 | _58.70_ | 34.07 | _32.25_ | 66.27 | 60.06 |
| PyTorch | 23.19 | 31.20 | 32.54 | 31.06 | 48.70 | 52.62 | 35.79 | 26.22 | 67.85 | _60.23_ |
| Adam-NSCL | _18.98_ | _23.53_ | _20.96_ | _31.57_ | _50.59_ | 52.95 | _35.98_ | 26.11 | **70.67** | 59.98 |
| **Q2N** | **17.79** | **25.77** | **20.01** | **34.30** | **57.11** | **62.40** | **36.84** | 33.26 | _69.75_ | **60.30** |

calibration), which indicate that our null space approximation method used in Q2N consistently outperforms the others. Notably, all null space based methods improve upon naive GPTQ, indirectly validating the importance of integrating null space optimization with LLMs quantization.

### 4.6 CLOSED-FORM SOLUTION VS BACKPROPAGATION (RQ5)

In Section 3.3, we theoretically derive a closed-form solution for the equivalent vector of the null space projection to satisfy practical inference. As known, the most intuitive way to get the optimal projection vector is learning via backpropagation. To compare the performance with our closed-form solution, we initialize a unit vector $m$ and optimize it via BP using the objective $\min \|(W - W_q) \times \Delta - (W - mW_q)\|_2^2$ with different training epochs (20, 50 and 100) and learning rates (5e-4, 1e-3 and 2e-3). As shown in Figure 3, the performance of BP-based projection vectors is unstable and lacks a consistent pattern. For example, 100 training epochs yield higher average accuracy, while 20 epochs result in better average perplexity. Under 20-epoch setting, increasing the learning rate leads to divergent trends in accuracy and perplexity. In contrast, our derived closed-form solution consistently performs the best, highlighting its superiority.

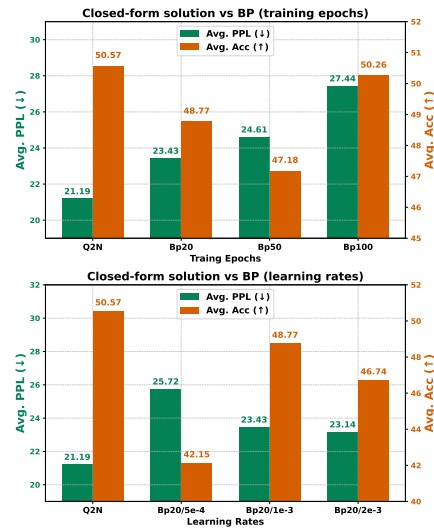

Figure 3: Performance comparison between our closed-form solution and BP (Up: training epochs; Down: learning rates) on LLaMA3.1-8B (3-bit, C4 Calib).

Taking all the discussion above into consideration, we prove that null space optimization can effectively further alleviate the quantization error. **Notably, we must emphasize that although the performance improvements of Q2N is relatively limited, it proves the feasibility of null space optimization. Compared to performance improvements at present, the novel perspective it provides for future research is more important.**

## 5 CONCLUSION

Existing PTQ methods suffer from inevitable quantization errors which also hinder the development of more advanced quantization algorithms. To provide a new direction for future research, in this paper we introduce null space optimization strategy into LLMs PTQ. We claim that by mapping the post-quantization weight perturbation into the null space of input activations, quantization errors can be effectively alleviated. By proposing an efficient and accurate example null space optimization method named Q2N and integrating it with several milestone baselines to validate the performance enhancements, we successfully prove the effectiveness of the idea of null space optimization for LLMs quantization. We hope our insightful perspective can provide fresh guideline for future quantization methods development.

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

## A    DETAILED PROOFS

In this section, we provide the detailed proofs corresponding to the null space projection approximation and our closed-form solution for the equivalent vector.

### A.1    PROOF FOR THE SHARED NULL SPACE OF $X$ AND ITS UNCENTERED COVARIANCE MATRIX $XX^T$

**Lemma A.1** *Given input activations $X \in \mathbb{R}^{n \times m}$. Then the null space of $X$ is equal to the null space of its uncentered covariance matrix $XX^T$, i.e.,*

$$Null(X) = Null(XX^T).$$

*Proof of Lemma A.1.* We show that $w \in \text{Null}(X)$ if and only if $w \in \text{Null}(XX^T)$.

($\Rightarrow$) Suppose $w \in \text{Null}(X)$. Then $wX = 0$, and thus

$$wXX^T = (wX)X^T = 0,$$

which implies $w \in \text{Null}(XX^T)$.

($\Leftarrow$) Conversely, suppose $w \in \text{Null}(XX^T)$. Then

$$0 = w(XX^T) = (wX)X^T$$

Since $w$ and $X$ are guaranteed to be nonzero, $wX = 0$ holds, *i.e.*, $w \in \text{Null}(X)$.

Therefore, $\text{Null}(X) = \text{Null}(XX^T)$.

### A.2    PROOF FOR NULL SPACE PROJECTION $\Delta = U_1 U_1^T$

**Lemma A.2** $\Delta = U_1 U_1^T$ *serves as the null space projection which can project the quantization numerical error of weights into the null space of $X$, i.e.,*

$$U_1 U_1^T X = \Delta X = (W - W_q)\Delta X = 0.$$

*Proof of Lemma A.2.* According to Section 3.2, the left singular vector of $XX^T$ is defined as $U = [U_2, U_1]$. We further define the singular values $\lambda = \begin{bmatrix} \lambda_2 & 0 \\ 0 & \lambda_1 \end{bmatrix}$, where all singular values of zero are in $\lambda_1$, *i.e.*, $\lambda_1 = 0$. Since $U$ is an orthogonal matrix, we can further derive that:

$$U_1^T XX^T = U_1^T U_2 \lambda_2 U_2^T = 0,$$

which indicates that the columns of $U_1$ span the null space for $XX^T$. According to (Meyer, 2023) (Eq. 5.13.4), the orthogonal projector of $XX^T$ can be elaborated as:

$$\Delta = U_1 U_1^T.$$

Thus $(W - W_q)\Delta X = (W - W_q)U_1 U_1^T X = 0$ holds.

Based on Lemma A.1 and Lemma A.2, we can get that $U_1 U_1^T$ serves as the null space projection of the input activation.

### A.3    THEORETICAL DERIVATION OF THE CLOSED-FORM SOLUTION FOR THE EQUIVALENT PROJECTION VECTOR

To satisfy practical inference constraints, the equivalent projection vector $\alpha$ must operate directly on the quantized weights ($W_q$) and achieve the same effect as the null space projection $\Delta$ applied to the post-quantization weight perturbation ($W - W_q$), so the objective function is formulated as:

$$\mathcal{L}(\alpha) = \|(W - W_q) \times \Delta - (W - \alpha W_q)\|_2^2 + \lambda(\alpha - 1)^2 I,$$

where the second term is the regularization term avoiding disturbing prior optimizations.

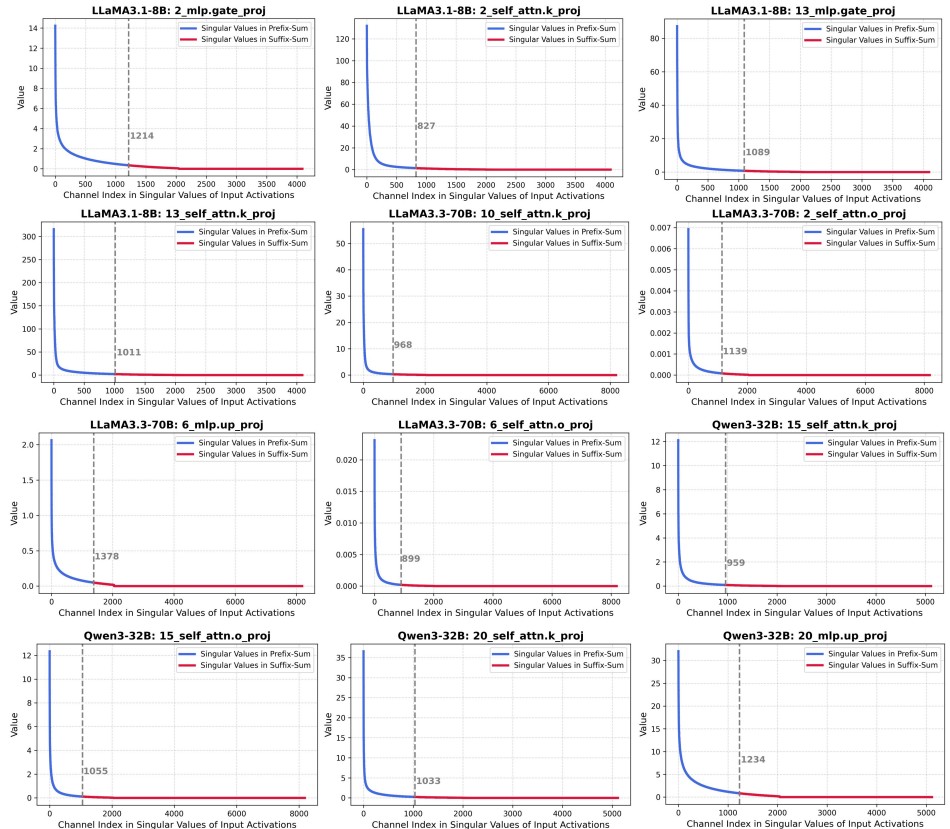

Figure 4: Segmentation results on singular values according to the Prefix-Suffix Sum Ratio.

Then we define $H = W - (W - W_q) \times \Delta$, take the derivative of $\mathcal{L}$ with respect to each dimension $i$ of $\alpha$ and set it to zero:

$$\frac{\partial \mathcal{L}}{\partial \alpha_i} = -2\langle W_q^i, H^i \rangle + 2\alpha_i \langle W_q^i, W_q^i \rangle + 2\lambda(\alpha_i - 1) = 0.$$

Rearranging the terms, we get:

$$\alpha_i(\langle W_q^i, W_q^i \rangle + \lambda) = \langle W_q^i, H^i \rangle + \lambda.$$

Solving for $\alpha_i$, we get the closed-form solution for the optimal equivalent projection vector $\alpha^*$:

$$\alpha_i^* = \frac{\langle W_q^i, H^i \rangle + \lambda}{\langle W_q^i, W_q^i \rangle + \lambda}.$$

Applying $\alpha^*$ to $W_q$, we can make the null space optimization for quantized model practically meaningful.

## B  VISUALIZATIONS OF THE PREFIX-SUFFIX SUM RATIO OF SINGULAR VALUES

We propose to use the Prefix-Suffix Sum Ratio of singular values to accurately approximate the null space projection. Here we present layer-wise visualizations of the segmentation results on several LLMs according to Eq. 6 with threshold $t = 0.1$ to highlight the outcomes. The visualizations are shown in Figure 5, where we remove the top 5 singular values to smooth the curves while not affecting the segmentation results.

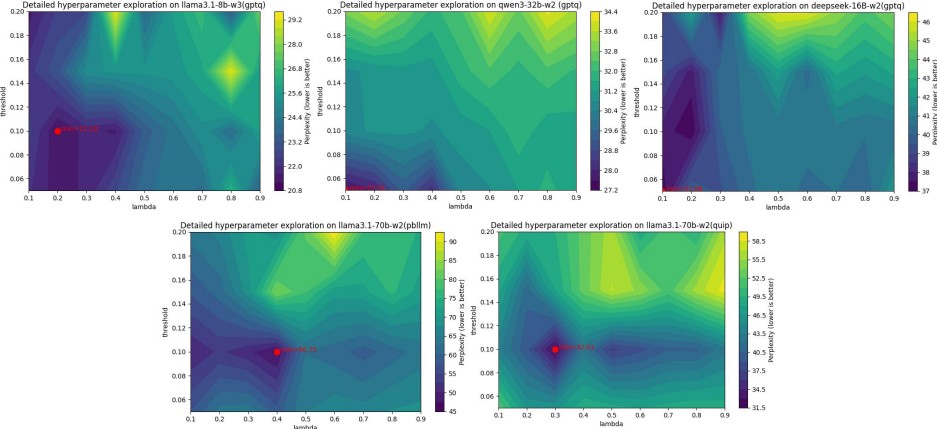

Figure 5: Hyper-parameter exploration based on performance on different LLMs and baselines.

| $t$ | $\lambda$ | C4 | | | WikiText2 | | |
|---|---|---|---|---|---|---|---|
| | | Wiki | PTB | C4 | Wiki | PTB | C4 |
| 0.1 | 0.1 | 21.46 | 27.15 | **18.85** | 15.86 | 29.43 | 23.77 |
| 0.1 | 0.2 | 17.79 | **25.77** | 20.01 | **12.31** | 28.61 | 23.16 |
| 0.1 | 0.3 | 17.92 | 30.34 | 21.00 | 14.95 | 34.12 | 23.83 |
| 0.1 | 0.4 | **17.50** | 25.90 | 21.11 | 14.50 | 33.89 | **22.92** |
| 0.1 | 0.5 | 21.84 | 35.63 | 24.62 | 15.50 | 34.74 | 26.71 |
| 0.1 | 0.6 | 24.80 | 30.95 | 25.74 | 12.69 | 34.70 | 23.63 |
| 0.1 | 0.7 | 18.93 | 31.38 | 24.92 | 15.99 | **27.24** | 27.73 |
| 0.1 | 0.8 | 21.19 | 26.38 | 24.39 | 13.47 | 29.90 | 23.49 |
| 0.1 | 0.9 | 19.04 | 34.01 | 23.99 | 12.71 | 29.49 | 23.98 |
| 0.05 | 0.2 | 20.05 | 23.50 | 20.89 | 13.04 | 37.72 | 23.46 |
| 0.15 | 0.2 | 20.80 | 33.55 | 23.30 | 14.49 | 30.76 | 24.36 |
| 0.2 | 0.2 | 18.90 | 25.03 | 20.19 | 15.32 | 31.73 | 25.25 |

Table 5: Hyper-parameters (ratio threshold $t$ and regularization coefficient $\lambda$) selection based on perplexity on LLaMA3.1-8B quantized by GPTQ with Q2N.

## C  ANALYSIS AND VISUALIZATION ON HYPER-PARAMETERS SELECTION

To achieve the best performance under our framework, we conduct a thorough investigation into the selection of the ratio threshold $t$ in null space approximation and the regularization coefficient $\lambda$ in closed-form solution of equivalent projection vector. Specifically, we employ coordinate descent to search for the optimal hyper-parameters. We first identify the optimal regularization coefficient $\lambda$ within the range $[0.1, 0.9]$ while fixing the ratio threshold $t = 0.1$, and then search for the optimal ratio threshold $t$ within $[0.05, 0.2]$. After research, we empirically give the overall optimal hyper-parameter combination in most cases: $t = 0.1$ and $\lambda = 0.2$. Table 5 gives the examples on LLaMA3.1-8B quantized by GPTQ (3-bit). In addition, we also discover some distinctions in specific baselines and models, such as LLaMA3-8B-GPTQ-WikiText2-3bit ($t = 0.1, \lambda = 0.3$) and LLaMA3.3-70B-LeanQuant-C4-3bit ($t = 0.05, \lambda = 0.2$). Therefore, we conduct extensive experiments across various models and baselines to identify a more robust and stable hyperparameter interval, with results shown in Figure. The heatmap illustrates that performance is generally better when $t < 0.1$ and $\lambda$ lies between 0.1 and 0.4. We also analyze why this range is better. As shown in Figure 5, when $t = 0.1$, the split point obtained by PSSR better fits the distribution of effective eigenvalues. When the threshold becomes larger, the split point shifts significantly into the long-tail region, resulting in poor estimation of the effective eigenvalues. When $\lambda$ is too large, the closed-form solution tends to keep the model weights unchanged rather than projecting the quantization error into the null space. In summary, we recommend searching for $t$ within $[0.1, 0.4]$ and selecting $\lambda$ from 0.05 and 0.1 based on our observations. We also experiment with defining the two hyperparameters as learnable for optimization, but discover that this approach underperforms compared to coordinate descent or grid search. We consider the manual hyper-parameter selection as a limitation of our current method.

| Model | Baselines | Calib | Q2N | Language Generation (↓) | | | Downstream Reasoning (%, ↑) | | | | | | |
|---|---|---|---|---|---|---|---|---|---|---|---|---|---|
| | | | | Wiki | PTB | C4 | ARC-c | ARC-e | HellaS | RACE | MMLU | PIQA | WinoG |
| LLaMA3-70B | QuIP | Wiki | × | 49.08 | 75.72 | 62.14 | 20.99 | 32.74 | **32.33** | 26.12 | 23.12 | 55.60 | 51.70 |
| | | Wiki | ✓ | **47.50** | **70.70** | **57.65** | **22.87** | **33.75** | 29.40 | **28.04** | **23.18** | **59.03** | **53.04** |
| | | C4 | × | 61.99 | 81.10 | 65.39 | 24.32 | 35.86 | 30.94 | 25.26 | 22.93 | 58.92 | **50.99** |
| | | C4 | ✓ | **54.78** | **68.64** | **58.55** | **24.74** | **38.01** | **32.87** | **27.37** | **24.43** | **59.30** | **50.99** |
| | PB-LLM | Wiki | × | 18.35 | 46.94 | 191.2 | 25.26 | 41.25 | 41.23 | 26.03 | 25.36 | 59.09 | 53.91 |
| | | Wiki | ✓ | **16.64** | **40.67** | **85.20** | **28.24** | **44.49** | **45.27** | **30.91** | **26.16** | **62.35** | **55.41** |
| | | C4 | × | 25.60 | 55.01 | 39.56 | 25.43 | **37.88** | 51.46 | 36.08 | 28.22 | 54.84 | 58.88 |
| | | C4 | ✓ | **22.37** | **41.31** | **36.07** | **26.71** | 30.13 | **55.22** | **37.42** | **29.58** | **61.21** | **60.93** |
| LLaMA3.1-70B | QuIP | Wiki | × | 42.73 | 71.39 | 52.85 | 22.70 | 32.41 | 32.74 | 27.85 | 23.12 | 55.71 | 51.38 |
| | | Wiki | ✓ | **23.73** | **43.66** | **35.51** | **25.60** | **35.73** | **38.86** | **29.28** | **23.75** | **59.47** | **55.33** |
| | | C4 | × | 46.04 | 62.78 | 50.21 | 23.21 | 36.15 | 33.35 | 26.41 | **23.40** | 58.43 | 48.54 |
| | | C4 | ✓ | **33.37** | **53.76** | **38.96** | **25.34** | **36.87** | **39.48** | **30.14** | 22.96 | **60.83** | **50.83** |
| | PB-LLM | Wiki | × | 25.04 | 74.17 | 521.5 | 25.94 | 43.43 | 46.16 | 29.38 | 24.17 | 61.59 | 55.49 |
| | | Wiki | ✓ | **22.66** | **60.74** | **444.7** | **27.73** | **46.63** | **47.53** | **29.47** | **24.33** | **63.44** | **56.75** |
| | | C4 | × | 34.73 | 70.51 | 104.9 | 21.59 | 29.38 | 49.79 | 32.73 | 27.96 | 53.21 | 57.93 |
| | | C4 | ✓ | **28.27** | **50.38** | **91.06** | **27.30** | **33.29** | **49.87** | **33.49** | **28.59** | **55.82** | **59.04** |
| LLaMA3.3-70B | QuIP | Wiki | × | 36.62 | 64.42 | 46.46 | 23.12 | 32.66 | 35.27 | 29.19 | 23.05 | 56.80 | 53.12 |
| | | Wiki | ✓ | **20.12** | **36.71** | **29.24** | **27.30** | **44.02** | **49.30** | **35.31** | **25.05** | **63.71** | **53.20** |
| | | C4 | × | 30.29 | 52.52 | 34.09 | 25.51 | 37.75 | 34.44 | 31.20 | 23.81 | 61.43 | 51.14 |
| | | C4 | ✓ | **24.04** | **38.66** | **28.85** | **27.05** | **45.03** | **49.47** | **34.55** | **24.98** | **66.43** | **52.88** |
| | PB-LLM | Wiki | × | 18.44 | 49.97 | 68.19 | 32.42 | 53.07 | 50.44 | 37.70 | 30.05 | 64.64 | 57.77 |
| | | Wiki | ✓ | **17.39** | **44.37** | **45.74** | **33.45** | **54.25** | **55.88** | **37.80** | **32.53** | **68.01** | **58.72** |
| | | C4 | × | 24.66 | 51.19 | 49.06 | 26.02 | 27.53 | 57.32 | 39.71 | 33.10 | 53.43 | 60.24 |
| | | C4 | ✓ | **23.16** | **45.57** | **39.98** | **27.47** | **28.16** | **57.32** | **40.19** | **33.96** | **54.62** | **61.17** |
| | OWQ | C4 | × | 82.47 | 57.45 | 49.74 | 48.89 | 72.90 | **52.50** | 23.92 | 62.78 | 76.44 | 64.48 |
| | | C4 | ✓ | **55.26** | **49.34** | **46.09** | **49.57** | **74.07** | 45.40 | **25.45** | **63.69** | **77.37** | **64.64** |
| | LeanQuant | C4 | × | 47.40 | 83.44 | 61.45 | 59.30 | 82.15 | 50.12 | 46.32 | 75.43 | 82.05 | 68.35 |
| | | C4 | ✓ | **40.18** | **66.07** | **53.63** | **59.56** | **83.25** | **60.80** | **47.27** | **76.30** | **82.37** | **69.61** |

Table 6: Detailed Evaluation results on different LLMs quantized by different weight-only baselines (with Q2N or not) on language generation and downstream reasoning tasks (2-bit).

| Model | Methods | Language Generation (↓) | | | Downstream Reasoning (%, ↑) | | | | | | |
|---|---|---|---|---|---|---|---|---|---|---|---|
| | | Wiki | PTB | C4 | ARC-c | ARC-e | HellaS | Lambda | MMLU | PIQA | WinoG |
| LLaMA3-8B | AWQ | 13.30 | 24.01 | 17.27 | 42.41 | 67.13 | 69.25 | 42.89 | 37.35 | 74.43 | 67.25 |
| | +Q2N | **12.29** | **20.98** | **16.16** | **43.42** | **68.07** | **70.06** | **43.98** | **38.44** | **75.27** | **68.65** |
| LLaMA3-70B | AWQ | 15.18 | 69.30 | 21.13 | 43.01 | 65.03 | 60.04 | 25.52 | 47.46 | 75.24 | 58.48 |
| | +Q2N | **13.96** | **55.24** | **19.82** | **44.34** | **66.07** | **60.95** | **26.54** | **48.42** | **76.08** | **59.64** |
| Qwen3-32B | AWQ | 11.84 | 20.06 | 14.66 | 45.8 | 70.71 | 75.91 | 41.41 | 69.68 | 49.51 | 60.69 |
| | +Q2N | **10.14** | **17.62** | **13.28** | **47.32** | **71.65** | **76.22** | **46.08** | **72.20** | **53.69** | **63.58** |

Table 7: Performance improvements when incorporating Q2N with AWQ (3-bit).

# D    DETAILED RESULTS WHEN INCORPORATING Q2N WITH OTHER BASELINES

In Section 4.3, we report the brief performance enhancements when incorporating our Q2N with three SOTA baselines within different strategies. Here we present the corresponding detailed performance on each metric in language generation and downstream reasoning tasks, where QuIP / PB-LLM / OWQ / LeanQuant are with Table 6, AWQ is shown in Table 7 and QuaRot is shown in Table 8.

| LLaMA | Methods | C4 | | | WikiText2 | | |
|---|---|---|---|---|---|---|---|
| | | Wiki | PTB | C4 | Wiki | PTB | C4 |
| 3-8B | QuaRot | 8.57 | 13.60 | 11.95 | **8.43** | 13.59 | 12.10 |
| | **+Q2N** | **8.51** | **13.41** | **11.88** | 8.45 | **13.56** | **12.02** |
| 3-70B | QuaRot | 72.15 | 313.02 | 166.41 | 108.25 | 264.18 | 342.07 |
| | **+Q2N** | **69.03** | **148.58** | **146.31** | **43.37** | **148.19** | **117.50** |
| 3.1-70B | QuaRot | 6.48 | 11.67 | 10.24 | 6.31 | 11.74 | 10.35 |
| | **+Q2N** | **6.39** | **11.55** | **10.19** | **6.27** | **11.48** | **10.21** |
| 3.3-70B | QuaRot | 7.39 | 13.48 | 11.54 | 7.38 | 13.54 | 11.86 |
| | **+Q2N** | **7.29** | **13.08** | **11.39** | **7.25** | **13.08** | **11.61** |

Table 8: Performance enhancement (PPL, ↓) when incorporating Q2N with QuaRot (W4A4).

