# OpenReview forum: "Boost Post-Training Quantization via Null Space Optimization for Large Language Models"
_ICLR.cc/2026/Conference — Submitted to ICLR 2026_

### Official Review · Reviewer_q2vZ · 2025-10-18

**Soundness:** 3
**Presentation:** 2
**Contribution:** 3
**Rating:** 4
**Confidence:** 4

**Summary:**

This paper introduces a novel post-training quantization (PTQ) optimization approach for large language models (LLMs) by leveraging null space theory, aiming to address the unavoidable quantization error in existing PTQ methods.
The core insight is that constraining the post-quantization weight perturbation $(W-W_q)$  to lie within the null space of input activations can effectively alleviate quantization error, as it minimizes the impact of such perturbation on the final output $(||(W-W_q)x||^2_2)\approx 0$. To validate this insight, the authors propose a framework with 3 key components: eplaces computationally expensive SVD with QR-based eigenvalue decomposition to handle large activation matrices in LLMs; Accurate null space Approximation with a threshold t, addressing the lack of exact zero singular values in LLM activations. Derives a vector $\alpha$ (with regularization coefficient $\lambda=0.2$ to integrate null space optimization.

**Strengths:**

1. Novel Theoretical Perspective: The paper introduces a fresh and meaningful direction for LLM post-training quantization (PTQ) by leveraging null space theory, addressing a key limitation of existing methods—their inability to mitigate the negative impact of unavoidable numerical errors beyond minimizing quantization error. This perspective, which links weight perturbation constraints to the null space of input activations, fills a gap in current quantization research and provides a new guideline for developing advanced quantization techniques.
2. The framework this paper proposed incorporates three well-designed components to ensure practicality for LLMs. These designs balance performance and efficiency, making the method applicable to real-world LLM inference scenarios. This paper not only introduces the null-space paradigm but also optimizes both speed and accuracy by addressing computational costs and the issue of zero singular values. These designs are orthogonal to existing quantization approaches and represent a plug-and-play new mechanism.
3. The paper conducts thorough experiments across diverse LLMs (e.g., LLaMA3, LLaMA3.1, Qwen2.5), multiple PTQ baselines (e.g., GPTQ, QuIP, LeanQuant), and different quantization scenarios (weight-only, weight-activation). Results consistently show improvements in both language generation (perplexity reduction) and downstream reasoning (zero-shot accuracy increase), demonstrating the method’s generality and effectiveness. Additionally, comparative experiments (e.g., vs. SVD, Adam-NSCL, backpropagation) further validate the advantages of the proposed components.

**Weaknesses:**

1. The paper acknowledges that the performance improvements brought by the framework are relatively moderate,  especially compared to some state-of-the-art PTQ methods that already achieve near-lossless results at lower bits. Additionally, the test sets mentioned in the paper are overly simplistic and focused exclusively on prefill-type tasks; it remains unclear how the method performs on generation-type tasks, such as GSM8K and HumanEval.

2. The framework relies on manual selection of key hyperparameters (e.g., the ratio threshold t for null space approximation and regularization coefficient $\lambda$ for the closed-form solution) via coordinate descent or grid search. The paper notes this as a limitation, as manual tuning not only increases the complexity of deployment but also may lead to suboptimal performance across different LLMs or quantization baselines, lacking an adaptive hyperparameter optimization mechanism.

3. The paper's approach primarily relies on estimating the null space of input samples, which inevitably makes it highly dependent on the quality of the calibration dataset. While it is undeniable that the method can achieve excellent performance by overfitting to a specific calibration set, its generalization to different scenarios remains questionable—the adaptability of the null space across diverse scenarios appears to be a fundamental challenge that cannot be easily overcome.

I’ll consider raising my score if you can solve my concerns.

**Questions:**

1. How can the inconsistency of null spaces across different calibration sets be addressed, so that the quantized model performs well not only on one or two specific tasks but generalizes across a broader range of tasks?

2. The notation in the paper is somewhat unclear. For example, in Section 3.2 ("Accurate Null Space Approximation"), both U1 and U2 appear simultaneously—what is the distinction between them, and what role does U2 play, especially since subsequent discussions primarily focus on U1?

3. Are the hyperparameters used in the paper generally applicable? If applied to different large language models, would this approach incur repeated hyperparameter tuning costs?

---

> ### Author Response · Authors · 2025-11-22
> **Response to Reviewer q2vZ (Part 1/2)**
>
> Dear Reviewers q2vZ,
>
> We really appreciate your reviews which enhance our rigor and persuasiveness. Regarding the questions you raised, we will provide answers below with the hope of offering you a better understanding.
>
> ---
> >**W1.** Moderate improvements and lack of results on GSM8K and humaneval.
>
> **A1.** The primary goal of our work is to introduce a new perspective for LLMs PTQ that null space optimization can be leveraged to alleviate the impact of quantization errors. Based on this idea, we present a feasible methods by proposing 3 useful techniques to further enhance current baselines. While the immediate performance improvements maybe modest, the consistent improvements across various tasks under low bit quantization demonstrate the effectiveness of this direction. Null space optimization **establishes a new conceptual foundation for future PTQ methods** to achieve even higher performance, and even potentially to eliminate the effect of quantization errors altogether, which is a more important innovation in our view .
>
> Following your suggestion, we provide GSM8K and HumanEval results under 3-bit on Qwen3-32B and LLaMA3-70Bl. As shown, clear performance improvements can be observed.
>
> |Model|bit|Method|gsm8k|humaneval|
> |:-|:-|:-|:-|:-|
> |Qwen3-32b|3|GPTQ|43.55|3.66|
> |||Q2N|**45.96**|**10.98**|
> ||4|GPTQ|67.10|35.98|
> |||Q2N|**70.41**|**37.20**|
> |llama3-70b|3|GPTQ|65.75|37.10|
> |||Q2N|**66.94**|**38.41**|
> ||3|AWQ|25.63|13.41|
> |||Q2N|**26.70**|**15.34**|
>
> We choose not to include 2-bit results because all evaluated baselines collapse on these two tasks at 2-bit so that offering no meaningful comparison. [1][2].
>
> >[1] Li S, Ning X, Wang L, et al. Evaluating quantized large language models[J]. arXiv preprint arXiv:2402.18158, 2024.
>
> >[2] Jin R, Du J, Huang W, et al. A comprehensive evaluation of quantization strategies for large language models[C]//Findings of the Association for Computational Linguistics ACL 2024. 2024: 12186-12215.
>
> ---
> >**W2.** Hyperparameters are manual selected
>
> **A2.** Adaptive hyperparameter selection is indeed an interesting idea, and we also attempt a greedy search strategy to determine the optimal hyperparameters during quantization. However, this approach significantly increases the computational cost.
>
> For this reason, we instead conduct extensive experiments across a wide range of model architectures and baselines to identify a stable and high-performing hyperparameter interval. **We visualize the results using a heatmap to identify suitable hyperparameter regions, please refer to Figure 5 in Appendix C.** From these results, we observe the best performance is achieved when t=0.05 to 0.1, and λ=0.1 to 0.4 on different LLMs and baselines, which means that our method is robust in such range.
> We also analyze why this range is better. As shown Figure 4 in Appendix B, when t=0.1, the split point obtained by PSSR better fits the distribution of effective eigenvalues. When the threshold becomes larger, the split point shifts significantly into the long-tail region, resulting in poor estimation of the effective eigenvalues. When λ is too large, the closed-form solution tends to keep the model weights unchanged rather than projecting the quantization error into the null space.
>
> ---
> >**W3.** The generalization of null space on diverse scenarios.
>
> **A3.** The calibration sets play important roles in PTQ, and we didn’t introduce additional calibration set for computing the null space but just **follow conventional PTQ methods** to use the most commonly adopted calibration sets WikiText and C4, which have consistently demonstrated strong generalization across downstream tasks and diverse application scenarios. For example, smoothquant which only use the two calibration sets to get smoothing parameters, also influenced by specific data, achieves excellent performance on various tasks .
>
> In addition, as shown in the table in our paper or below, our null space optimization method derived from these two calibration sets **improves quantization performance on different types of tasks**, including language generation and various downstream tasks such as commonsense reasoning.
>
> |Model|Q2N|arc-c|arc-e|hella|mmlu|piqa|wino|wiki|c4|
> |:-|:-|:-|:-|:-|:-|:-|:-|:-|:-|
> |llama3-8b|✘|24.66|40.03|48.72|29.93|60.01|62.04|19.83|23.08|
> ||✔|**28.92**|**45.20**|**62.11**|**32.09**|**63.76**|**62.27**|**17.55**|**20.20**|
> |deepseek-16b|✘|23.81|35.40|38.93|23.49|60.34|50.59|48.45|48.34|
> ||✔|**26.96**|**40.32**|**41.77**|**29.85**|**24.34**|**62.08**|**51.85**|**45.17**|**35.23**|
> |qwen3-32b|✘|128|25.43|32.45|47.85|24.88|57.5|51.46|37.1|26.59|
> ||✔|**28.5**|**35.73**|**53.09**|**25.24**|**59.85**|**53.59**|**26.98**|**22.72**|
>
> ---

---

> ### Author Response · Authors · 2025-11-22
> **Response to Reviewer q2vZ (Part2/2)**
>
> >**Q1.** How to ensure the generalization.
>
> **AQ1.** As described in A3, our null space optimization method follows conventional PTQ methods that use Wikitext2 and C4 as calibration set can consistently improve the performance of quantized LLMs on language generation and various downstream tasks.
>
> ---
> >**Q2.** What role does U2 play?
>
> **AQ2.** As in line 164-165, U2 is the remaining submatrix in the left singular values that corresponds to the principle components (or none zero singulars), which means that XX^T=U2λU2, while U1 corresponds to the zero singulars. In null space approximation, we will not use U2 but only U1.
>
> ---
> >**Q3.** Generalization of hyperparameters.
>
> **AQ3.** As our answers in A2, we conduct extensive experiments across diverse llms with different architectures and baselines to identify a stable hyperparameter range. We provide this range as our recommended configuration for reproduction, and the followers can achieve satisfactory performance as long as the hyperparameters are selected within this interval.
>
> ---
> Thank you again for your reviews! It will be encouraging that you can raise your score if we have addressed all your issues. And we are also very happy to further discuss with you.
>
> Best,
>
> Authors

---

> > ### Comment · Reviewer_q2vZ · 2025-11-23
> >
> > Your response still feels somewhat vague overall. Indeed, null space quantization is an extremely novel direction, but the experimental results presented in the paper do not deliver amazing outcomes—this is a major point of hesitation for me. Additionally, this method relies on a calibration dataset. It appears that the authors merely listed some conventional experimental results without providing a theoretical explanation for how to address the generalization challenge. Clearly, Wikitext-2 and C4 cannot cover certain reasoning and question-answering scenarios, and the model's performance in "chain-of-thought" reasoning is more or less compromised. In summary, while the novelty of this paper is certainly worthy of recognition, its practical contributions and actual performance leave me quite hesitant.

---

> > > ### Author Response · Authors · 2025-11-26
> > > **Response to Reviewer q2vZ R2 (Part1/2)**
> > >
> > > Dear Reviewer q2vZ,
> > >
> > > Thank you very much for the timely discussion. We really apologize for the delay in our response, as running COT experiments for the rebuttal require lots of time.
> > >
> > > >**RQ1.** Little improvements
> > >
> > > **A1.** Our method is an enhancement for existing baselines, all of which optimize quantization by minimizing numerical quantization error and have already **reached optimal performance under their corresponding paradigm**. The fact that our null space optimization can **further improve these well-converged baselines** demonstrates that null space optimization enables **further convergence beyond what numerical-error minimization alone can achieve**. This highlights the potential of combining null space optimization with existing numerical-error based methods, or even pure null space optimization in future quantization research.
> > >
> > > Regarding your comment that our improvements are small, we would like to emphasize that null space optimization **achieves notable improvements in some cases where numerical-error-based methods perform poorly**. As shown in the table below (C4 calib), our approach achieves notable improvements that existing numerical error-only baselines cannot obtain.
> > >
> > > |Model|Q2N|wiki|ptb|c4|
> > > |:-|:-|:-|:-|:-|
> > > |llama3.1-8b|✘|24.55|36.72|26.79|
> > > ||✔|17.79(**-6.76**)|25.77(**-10.95**)|20.01(**-6.78**)|
> > > |DeepSeek|✘|48.45|-|48.34|
> > > ||✔|45.17(**-3.28**)|-|35.23(**-13.11**)|
> > > |Qwen3-32B|✘|37.10|54.97|26.59|
> > > ||✔|26.98(**-10.12**)|42.97(**-12.00**)|22.72(**-3.87**)|

---

> > > ### Author Response · Authors · 2025-11-26
> > > **Response to Reviewer q2vZ R2 (Part2/2)**
> > >
> > > >**RQ2.** Null space based on calibration set will harm generalization and performance on COT reasoning.
> > >
> > > **A2.**  We would like to clarify that the null space projection matrices obtained from different calibration sets are **highly similar**. This indicates that our null space optimization is **not sensitive to the choice of calibration data** and will not undermine the model’s generalization ability.
> > >
> > > First, the activation matrices are high-dimensional but low-rank, with ranks typically in the single digits. **The statistical differences across small-scale calibration sets are quite small**. As a result, the covariance matrices computed from different calibration sets remain extremely similar. Consequently, the null space bases estimated from different calibration sets correspond merely to a little different orthonormal representations of the **similar underlying low-dimensional subspace** [1][2][3], rather than fundamentally quite different subspaces.
> > >
> > > >[1] Aghajanyan A, Gupta S, Zettlemoyer L. Intrinsic dimensionality explains the effectiveness of language model fine-tuning[C]//Proceedings of the 59th annual meeting of the association for computational linguistics (volume 1: long papers). 2021: 7319-7328.
> > >
> > > >[2] Ethayarajh K. How contextual are contextualized word representations? Comparing the geometry of BERT, ELMo, and GPT-2 embeddings[J]. arXiv preprint arXiv:1909.00512, 2019.
> > >
> > > >[3] Mu J, Andreas J. Compositional explanations of neurons[J]. Advances in Neural Information Processing Systems, 2020, 33: 17153-17163.
> > >
> > > To support this observation, we measure the Grassmannian Distance of the null space projection matrices computed using three math reasoning datasets, GSM8K,SVAMP and MAWPS, as the calibration sets across all layers. As shown in the table below, **the difference between these matrices and those obtained using C4 are around 1e-5, which is effectively negligible.**
> > >
> > > |Calib set|q|k|v|o|gate|up|down|
> > > |:-|:-|:-|:-|:-|:-|:-|:-|
> > > |GSM8K|2.379e-5|2.379e-5|2.379e-5|3.563e-5|2.608e-5|2.608e-5|7.051e-5|
> > > |SVAMP|2.393e-5|2.393e-5|2.393e-5|3.571e-5|2.627e-5|2.627e-5|7.127e-5|
> > > |MAWPS|2.380e-5|2.380e-5|2.380e-5|3.563e-5|2.611e-5|2.611e-5|7.062e-5|
> > >
> > > In addition, you mention that potential generalization issues might harm performance on CoT tasks. To verify this, we conduct experiments on Qwen3-32B on four different CoT benchmarks， gsm8k, gpqa, bbh and mgsm. The results show that our method **still achieves improvements over prior baselines on all these CoT tasks**. This further demonstrates that our approach does not compromise, and in fact enhances, CoT reasoning performance.
> > >
> > > |bits|gsm8k-cot|gpqa-cot|bbh-cot|mgsm-cot|
> > > |:-|:-|:-|:-|:-|
> > > |3|47.42|48.93|10.95|12.10|
> > > ||**48.60**|**11.83**|**50.07**|**12.80**|
> > > |4|71.11|53.14|14.51|52.40|
> > > ||**72.07**|**53.58**|**15.39**|**53.30**|
> > >
> > > Furthermore, we also use GSM8K as the calibration set and evaluate the performance on GSM8K compared with C4 calibration on Qwen3-32B. The results below show a similar performance and GSM8K-calibrated even performs slightly worse than the C4-calibrated. More importantly, in both settings null space optimization consistently improves performance. These findings not only confirm that C4 is a better selection for PTQ, but also further demonstrate that our method **will not introduce any generalization issues**.
> > >
> > > |bits|calib|q2n|GSM8K|
> > > |:-|:-|:-|:-|
> > > |3|GSM8K|✘|40.56|
> > > |||✔|42.68|
> > > ||C4|✘|**43.55**|
> > > |||✔|**45.96**|
> > > |4|GSM8K|✘|66.41|
> > > |||✔|68.20|
> > > ||C4|✘|**67.10**|
> > > |||✔|**70.41**|
> > >
> > > Finally, we would like to emphasize that our method does **not learn a compensation model using activation statistics**. Instead, we use the activation null space to absorb the quantization error of the weights. As long as the null spaces derived from different calibration sets are **similar**, which our measurements confirm, the generalization capability of the quantized model will **not be affected**.
> > >
> > > ---
> > >
> > > Thanks again for your careful review and on-time discussion. If you have any other concerns, please feel free to contact with us. And it will be encouraging that you can raise your score if we have addressed your issues.
> > >
> > > Best regards,
> > > Authors

---

> > > > ### Comment · Reviewer_q2vZ · 2025-11-26
> > > >
> > > > I cannot agree with the authors' analysis on the calibration dataset.
> > > >
> > > > Based on my observations, there must be significant statistical differences between different calibration datasets. I am puzzled by how the authors measured the covariance of different calibration datasets as extremely similar—such a finding contradicts empirical evidence. To adopt an alternative perspective: if the covariance of different calibration datasets were indeed similar, methods like AWQ and GPTQ would not exhibit varying performance across different calibration datasets. Taking this to an extreme, would calibration using random numbers also be feasible? Therefore, I argue that null-space quantization is necessarily strongly correlated with the distribution of the calibration dataset rather than being agnostic to it.
> > > >
> > > > Notably, the authors' own experimental comparisons demonstrate that calibration on the C4 dataset yields the optimal results, indicating that C4 possesses distinct distributional advantages not shared by GSM8K or other mathematical datasets.
> > > >
> > > > Consequently, I find the authors' response to this issue unpersuasive.

---

> ### Author Response · Authors · 2025-11-26
> **Response to Reviewer q2vZ R3**
>
> Dear Reviewer q2vZ,
>
> We apologize for the confusion caused by our earlier explanation, and we fully agree with your observation regarding the distribution shift across calibration sets which is consistent with our initial rebuttal. The Grassmannian Distance reported in our previous table is intended to measure the overlap between the low-dimensional subspaces of the covariance matrices at each layer. When using only a small calibration set, the gap between these subspaces indeed appears limited. When evaluating MSE difference with c4, the difference becomes larger, as below, which is consistent with your expectation.
>
> |Calib|q|k|v|o|gate|up|down|
> |:-|:-|:-|:-|:-|:-|:-|:-|
> |Wiki|12.97|12.97|12.97|27.74|19.45|19.45|43.64|
> |GSM8K|19.23|19.23|19.23|28.79|23.68|23.68|38.80|
>
> As we mention in our initial rebuttal, **calibration set selection is indeed crucial**. To further illustrate this point, we also provide results using randomly generated data (random alphanumeric characters) for calibration. As shown in the table below, **the performance degrades significantly when using random data**, confirming the importance of selecting an appropriate calibration set.
>
> |Calib|Gsm8k|wiki|ptb|c4|
> |:-|:-|:-|:-|:-|
> |Random|1.36|37.09|54.50|36.11|
> |Gsm8k|40.56|23.78|34.81|22.76|
> |C4|**43.55**|**11.55**|**18.61**|**13.95**|
>
> *So why we use C4 as calibration set, and whether select a task-specific dataset such as GSM8K for calibration and null space optimiziation will perform better and not cause generalization problem?* As shown in our previous rebuttal, using C4 for calibration actually achieves better performance on GSM8K than calibrating directly on GSM8K itself. This phenomenon arises because quantization and null space optimization depend **not only on the calibration data, but more fundamentally on the model’s pretraining distribution.** Common calibration sets such as **C4 and WikiText2 better match the pretraining corpora of LLMs, so they are more suitable for PTQ and will cover the inherent performance and generalization of pretrained model.** For example, RedPajama-1T[1], which is the corpus used for LLMs pretraining like LLaMA and Qwen, includes C4 and Wikipedia. This is precisely why nearly all PTQ methods select C4 and WikiText2 as their standard calibration datasets. Since the pretrained model has never seen task-specific data such as GSM8K, calibrating and computing the null space directly on GSM8K introduces a distributional gap relative to the model’s original pretraining data. **This mismatch disrupts the pretrained knowledge structure and inherent capabilities, leading to worse performance even on theirselves than using C4 calibration.** Thus, using downstream-task data for calibration or null space optimization does not guarantee better performance on that task but does harm. *In many cases, it may require additional finetuning [2] or test-time adaptation [3] to be effective, which is outside the scope of our work.* **In contrast, using the dataset that matches the pretraining distribution (e.g., C4 or WikiText) for calibration or null space optimization better preserves the pretrained model’s inherent knowledge structure and covers its generalization ability across reasoning and generation tasks.** We will add this explanation to our next version.
>
> >[1] Weber M, Fu D, Anthony Q, et al. Redpajama: an open dataset for training large language models[J]. Advances in neural information processing systems, 2024, 37: 116462-116492.
>
> >[2] Li Y, Yu Y, Liang C, et al. Loftq: Lora-fine-tuning-aware quantization for large language models[J]. arXiv preprint arXiv:2310.08659, 2023.
>
> >[3] Zhao J, Zhang M, Zeng C, et al. LRQuant: Learnable and robust post-training quantization for large language models[C]//Proceedings of the 62nd Annual Meeting of the Association for Computational Linguistics (Volume 1: Long Papers). 2024: 2240-2255.
>
> ---
> We have done our best to address each point and clarify any misunderstandings. We are very happy to provide any additional clarifications that you may need.
>
> Best,
>
> Authors

---

### Official Review · Reviewer_fcAe · 2025-10-30

**Soundness:** 2
**Presentation:** 3
**Contribution:** 3
**Rating:** 6
**Confidence:** 5

**Summary:**

The paper “Boost Post-Training Quantization via Null Space Optimization for Large Language Models” presents a post-training quantization (PTQ) correction method designed to mitigate quantization errors in large language models. The proposed approach identifies the null space of activations for each layer and formulates an optimization problem that approximates this null-space projection using a vector–matrix Hadamard product.

**Strengths:**

- The proposed solution has the original element of applying the null-space correction to the quantization error.
- The proposed quantization correction can be seamlessly integrated into existing quantized inference pipelines.
- It introduces no additional latency or memory overhead since it can be merged into the per-channel scaling vector.
- Experimental results demonstrate consistent and significant improvements across various quantization methods, supporting the effectiveness and generality of the approach.

**Weaknesses:**

- The relationship between eigenvalue decomposition and singular value decomposition of the covariance matrix is well established and widely used. Therefore, the “efficient eigenvalue decomposition” component should not be considered a novelty contribution.
- It would strengthen the paper to include evaluations on quantization methods that explicitly consider activation statistics, such as AWQ. It is not clear, whether such methods can benefit from the proposed approach.
- The WA quantization results appear unimpressive. Since the perplexity metric is highly sensitive, the results in Table 2 do not convincingly indicate meaningful improvements. The only notable exception is the LLaMA3-70B QuaRot result in Table 7, which shows a substantial difference. This raises the question of whether QuaRot was applied correctly in that case.
- There are a list of inconsistencies or logical errors throughout the paper that hinder the paper clarity:
- a. Since the covariance matrix is positive semi-definite, its eigenvalues are non-negative. Therefore, taking the absolute value of eigenvalues (lines 177–178) is unnecessary.
- b. The equation in line 668 is invalid when w is not a scalar. The proof should adopt a different formulation.
- c. If a is a vector, the regularization term (a−1) should be expressed as an l2 norm.
- d. The Hadamard product in Equation (8) should be explicitly denoted to avoid ambiguity.
- e. Figure 3 would be more interpretable as a bar plot or an alternative visualization.


The paper presents strong experimental results and a well-executed quantization correction method; however, the underlying idea offers limited originality, bearing resemblance to existing activation-aware quantization approaches. The use of null-space optimization introduces a degree of novelty, but overall, the contribution feels somewhat incremental for a full-length conference paper. The evaluation could be strengthened by including comparisons with methods such as AWQ to better demonstrate general applicability. Additionally, refining the approach for WA quantization and improving the clarity of exposition, as noted in the comments, would enhance the paper’s quality and impact.

**Questions:**

- Can you provide the impact of your method with activation-aware quantization approaches like AWQ or smooth quant?
- When comparing the closed-form and BP solutions, the optimization objectives differ. Specifically, the BP objective (line 458) omits the regularization term present in Equation (8). Please clarify this discrepancy.
- How many samples from the Wiki and C4 calibration datasets were used to compute the null space of X? Table 3 indicates that computing the null space for a single transformer block takes approximately 4 seconds—please specify how many samples were used for that estimation.
- Please, correct the errors pointed out in the "Weaknesses" section.

---

> ### Author Response · Authors · 2025-11-22
> **Response to Reviewer fcAe**
>
> Dear Reviewer fcAe,
>
> Thanks for your positive and insightful reviews which help us strengthen the manuscript. We have provided answers below with the hope of offering a better understanding.
>
> ---
> >**W1.** Efficient eigenvalue decomposition is not a contribution.
>
> **A1.** In LLM quantization, SVD-based methods, such as ASVD and SVDLLM, have been widely used. However, these methods overlook the extremely high dimensionality of weights and activations in LLMs **makes SVD computationally prohibitive**, where efficiency is essential for both research and deployment. As shown in the table, we observe that SVD decomposition introduces substantial latency that slows down the runtime significantly. In addition, our firstly introduced eigenvalue decomposition reduces the time required to get the null space projection by **tens of times**, so our approach is far more suitable for practical LLM quantization.
>
> |Method|Q|K|V|O|Up|Gate|Down|
> |:-|:-|:-|:-|:-|:-|:-|:-|
> |SVD|4.14|4.15|4.14|3.16|3.48|3.54|142.84|
> |Ours|**0.15**|**0.17**|**0.15**|**0.15**|**0.16**|**0.15**|**3.35**|
>
> ---
> >**W2.** Provide performance on AWQ
>
> **A2.** Thank you for your suggestion. We provide the performance of our method on AWQ on two widely used LLMs family, Llama3 and QWen3. As the following table shown, consistent performance improvements can be consistently observed. We did not include 2-bit experiments because AWQ consistently collapses under 2-bit quantization regardless of groupsize=128. This behavior has also been documented in the following two papers. We have followed your comment to include the evaluations in our new version, please refer to **Table 7 in Appendix D**.
>
> |Model|Method|Wiki|ptb|c4|arc-c|arc-e|hellas|lambada|mmlu|piqa|wino|
> |:-|:-|:-|:-|:-|:-|:-|:-|:-|:-|:-|:-|
> |qwen3-32b|AWQ|11.84|20.06|14.66|45.8|70.71|75.91|41.41|69.68|49.51|60.69|
> ||q2n|**10.14**|**17.62**|**13.28**|**47.32**|**71.65**|**76.22**|**46.08**|**72.2**|**53.69**|**63.58**|
> |llama3-8b|AWQ|13.3|24.01|17.27|42.41|67.13|69.25|42.89|37.35|74.43|67.25|
> ||q2n|**12.29**|**20.98**|**16.16**|**43.42**|**68.07**|**70.06**|**43.98**|**38.44**|**75.27**|**68.65**|
> |llama3-70b|AWQ|15.18|69.30|21.13|43.01|65.03|60.04|25.52|47.46|75.24|58.48|
> ||q2n|**13.96**|**55.24**|**19.82**|**44.34**|**66.07**|**60.95**|**26.54**|**48.42**|**76.08**|**59.64**|
>
> ---
> >**W3.** WA performance appear unimpressive.
>
> **A3.** Quarot is currently the SOTA method for WA quantization, whose performance at W4A4 is already **very close to FP16**, so there is only limited room for further improvement. Nevertheless, our method achieves further improvements based on Quarot which demonstrates its effectiveness. The larger improvement on LLaMA3-70B is due to the fact that this model contains **large amounts of outliers in weight matrices**, whcih leads to significant performance degradation[1][2], so that our method demonstrates a meaningful improvement based on the worse initial performance.
>
> A similar phenomenon can be observed on SmoothQuant. As shown in the table below, SmoothQuant experiences severe degradation at W4A4 on llama3-8b, whereas our method substantially boosts its performance. We didn’t include W8A8 because smoothquant is nearly lossless. Although moderate improvements when at higher-bit/performance, it is sufficient to validate the robustness and practical value of our approach in challenging low-bit scenarios.
>
> |Method|wiki|ptb|c4|
> |:-|:-|:-|:-|
> |Smooth|4.3e3|4.0e3|3.6e3|
> |q2n|**167.63**|**271.83**|**160.44**|
>
> >[1] Qin M. The uniqueness of llama3-70b series with per-channel quantization[J]. arXiv preprint arXiv:2408.15301, 2024.
> >[2] Huang W, Zheng X, Ma X, et al. An empirical study of llama3 quantization: From llms to mllms[J]. Visual Intelligence, 2024, 2(1): 36.
>
> ---
> >**W4.** Inconsistency and logical errors
>
> >**A4.** We really appreciate your careful reviews. We have corrected the inconsistencies and logical errors in our new version PDF
>
> ---
> >**Q1.** Provide results on activation-aware quantization approaches like AWQ or smoothquant.
>
> **AQ1.** Thank you for your suggestion. Please refer to A2 and A3, we achieve consistent performance improvements on 3-bit AWQ on various tasks and meaningful results on W4A4 SmoothQuant. We have added the results in our new version PDF.
>
> ---
> >**Q2.** Clarify the BP objective omits the regularization term.
>
> **AQ2.** In our comparison, BP method selects the widely used AdamW optimizer, which inherently includes a regularization term. Therefore, we do not add an additional regularization term to the optimization objective term.
>
> ---
> >**Q3.** How many samples are used to compute null space.
>
> **AQ3.** We randomly collect 128 samples with 2048 tokens from Wiki and C4 for computing the null space projection.
>
> ---
> Thank you again for your reviews! It will be encouraging that you can raise your score if we have addressed all your issues. And we are also very happy to further discuss with you.
>
> Best,
>
> Authors

---

> ### Author Response · Authors · 2025-11-26
> **A kind reminder of discussion : We would like to learn Reviewer’s opinion and address any remaining concerns.**
>
> Dear Reviewer fcAe,
>
> We would kindly like to inquire if you would get a chance to review our response and if there are any remaining questions we can address.
>
> Your insights, both the constructive suggestions and areas of contention, have been crucial for us. We have done our best to address each point and clarify any misunderstandings. We are truly keen to have a constructive dialogue with you to refine our work further.
>
> Best regards,
>
> Authors

---

### Official Review · Reviewer_nHoy · 2025-10-31

**Soundness:** 3
**Presentation:** 3
**Contribution:** 2
**Rating:** 4
**Confidence:** 4

**Summary:**

This paper proposes a novel perspective on post-training quantization (PTQ) in LLM: constraining the quantization error to the null space of the input activations to mitigate error propagation. The authors present an exemplary implementation—Q2N (Quantize-to-Nullspace)—combined with baselines such as GPTQ, QuIP, LeanQuant, and QuaRot, demonstrating consistent but limited improvements in language generation and downstream tasks.

Overall, this is an interesting mathematical perspective, but the method itself is rather empirical, lacking theoretical depth and experimental persuasiveness, making it a typical "concept-first but engineering-light" paper. It's more of an inspirational exploration than a technology that can truly change the landscape of quantization.

**Strengths:**

1. Introducing null space theory into quantization error analysis is indeed a first in LLM quantization literature (previously it was mostly used in LoRA-Null, AlphaEdit, etc.), and it is quite inspiring.
2. The core lemma (if the error is in the input null space, the output error approaches zero) is logically rigorous and its derivation is complete.
3. Complete derivation, pseudocode, and open-source links are provided, along with engineering execution specifications.
4. Compatibility has been verified through testing on multiple frameworks, including GPTQ, QuIP, LeanQuant, and QuaRot.

**Weaknesses:**

### **1. Limited contribution and marginal improvements**
While the proposed Null Space optimization approach is novel, its methodology is overly simplistic, merely adding an approximate projection operation to existing frameworks such as GPTQ. Experimental results show limited performance improvement—only 1–3 percentage points for most tasks, and no significant improvement for some metrics.

### **2. Over-simplified projection mechanism**
The authors claim to reduce error propagation by projecting the quantization error (W − Wp)  onto the null space of the input activation. However, in the actual implementation, only a channel-level vector α ∈ ℝ^{C_out} is introduced to replace the entire projection matrix. This substitution is overly simplistic, equivalent to multiplying the output dimension by a scaling factor. It fails to effectively model the high-dimensional structure of the null space and cannot truly guarantee  **(W − Wp)X ≈ 0**.  Therefore, this implementation is more like a "numerical correction" than a true projection optimization.
### **3. Lack of ablation and mechanism analysis**
The paper fails to analyze which component actually contributes to the reported improvements.
Multiple elements could influence results—(a) replacing SVD with eigen decomposition, (b) the prefix-suffix ratio for null-space approximation, and (c) the regularization term on **α**—yet none are independently evaluated.
Both the threshold **t = 0.1** and regularization weight **λ = 0.2** are empirically chosen without justification or sensitivity study, leaving questions about robustness and reproducibility.

**Questions:**

Please see weaknesses.

---

> ### Author Response · Authors · 2025-11-22
> **Response to Reviewer nHoy (Part1/2)**
>
> Dear Reviewer nHoy，
>
> We really appreciate your reviews which enhance our rigor and persuasiveness. Regarding the questions you raised, we will provide answers below with the hope of offering you a better understanding.
>
> ---
> >**W1.** Limited contribution and marginal improvements
>
> **A1.** Although we have certain limitations, Our method show consistent performance improvements across many mainstream baselines (e.g., GPTQ, QuIP, Quarot), a range of advanced LLMs (e.g., Llama3, DeepSeek, Qwen3), and multiple downstream tasks, especially for ultra low-bit large groupsize quantization, our method achieves more than 50% improvements. More importantly, we introduce **a new paradigm that leveraging null space optimization to enhance quantization which provide a fresh view**. This contribution goes beyond the immediate empirical gains and highlights the broader potential impact of our approach.
>
> ---
> >**W2.** Over-simplified projection mechanism
>
> **A2.** The reason we apply only an equivalent vector instead of a full projection matrix is that we consider the practical deployment of LLMs. There are only quantized weights and no their FP counterparts storing during inference, so we **cannot directly apply the full precision matrix to the quantization error.** In addition, storing a complete projection matrix for each weight matrix would effectively **double the model size**, resulting in prohibitive overhead. In contrast, using an equivalent vector incurs negligible additional cost, and as shown in the table, it achieves nearly the same performance as the full matrix (we load an extra full precision model to satisfy the projection matrix applying on quantization error directly ). Therefore, storing only an equivalent vector is a far more practical solution for LLMs.
>
> |Model|Method|wiki|c4|Arc-c|Arc-e|HellaS|MMLU|PIQA|Wino|
> |:-|:-|:-|:-|:-|:-|:-|:-|:-|:-|
> |llama3-8b|matrix|16.94|18.73|30.41|44.78|62.75|33.86|65.06|64.43|
> ||vector|17.55|20.20|28.92|45.20|62.11|32.09|63.76|62.27|
> |deepseek|matrix|44.82|34.65|40.97|42.36|30.31|25.40|62.39|52.66|
> ||vector|45.17|35.23|26.96|40.32|41.77|29.85|24.34|62.08|51.85|
> |qwen3-32b|matrix|26.01|22.14|29.25|36.28|53.98|25.62|60.12|54.24|
> ||vector|26.98|22.72|28.50|35.73|53.09|25.24|59.85|53.59|
>
> Additionally, you mention that our method may have difficulty ensuring (W−Wp)X=0. We would like to clarify that the performance improvements indicate our method **successfully project part of the quantization error into the null space**, which already breaks the limitations of prior methods and get further improvements. This can also be proved by the Euclidean distance of FP and quantized outputs.
>
> |Q2N|q|k|v|o|gate|up|down|
> |:-|:-|:-|:-|:-|:-|:-|:-|
> |✘|14.94|10.69|1.87|0.41|12.00|7.47|13.07|
> |✔|**8.89**|**6.27**|**1.05**|**0.13**|**7.89**|**3.65**|**8.14**|
>
> ---

---

> ### Author Response · Authors · 2025-11-22
> **Response to Reviewer nHoy (Part2/2)**
>
> >**W3.** Lack of ablation and mechanism analysis
>
> **A3.** Thank you for your comprehensive comment. Our three design components are each introduced to address a distinct and essential aspect of null-space optimization.
>
> (a)*Replacing SVD with eigenvalue decomposition.* This is designed for **efficiency improvement**. As the runtime on llama3-8b shown in the table below, this substitution obtains computation speedups of several orders of magnitude.
>
> |Method|Q|K|V|O|Up|Gate|Down|
> |:-|:-|:-|:-|:-|:-|:-|:-|
> |SVD|4.14|4.15|4.14|3.16|3.48|3.54|142.84|
> |Ours|**0.15**|**0.17**|**0.15**|**0.15**|**0.16**|**0.15**|**3.35**|
>
> (b)*PSSR for accurate null-space approximation.* The PSSR module is central to achieve null space projection, which **directly determines the performance**. As the results shown, our PSSR-based approximation significantly outperforms prior null space optimization baselines.
>
> |Method|wiki|c4|Arc-c|Arc-e|HellaS|MMLU|PIQA|Wino|
> |:-|:-|:-|:-|:-|:-|:-|:-|:-|
> |GPTQ|24.55|26.79|31.23|49.71|58.70|32.25|66.27|60.06|
> |PyTorch|23.19|32.54|31.06|48.70|52.62|26.22|67.85|60.23|
> |Adam-NSCL|18.98|20.96|31.57|50.59|52.95|26.11|70.67|59.98
> |Ours|**17.79**|**20.01**|**34.30**|**57.11**|**62.40**|**33.26**|**69.75**|**60.30**|
>
> (c) *A regularization term for computing the equivalent vector.* This design ensures that the final equivalent vector which is suitable for practical deployment can be computed in closed form with **numerical stability**. It is also important for performance improvement. As the table in A2 shown.
>
> Regarding the concern about our hyperparameter selection, we have provided a  optimal and stable hyperparameter selection range in Appendix C which is determined through quite extensive experiments on Llama, Qwen, and DeepSeek using multiple baselines. The search range for λ is 0.1 to 0.9, and t ranges from 0.05 to 0.2. **We visualize the results using a heatmap to identify stable and optimal hyperparameter regions, please refer to Figure 5 in Appendix C in our new version PDF**.  We find the optimal choice of λ falls within the range of 0.1 to 0.4 and t is 0.05 to 1, which means that our method is robust in such range. Therefore, future followers only need to select hyperparameters within this range, and the robustness of our method will not be affected.
>
> ---
> Thank you again for your reviews! It will be encouraging that you can raise your score if we have addressed all your issues. And we are also very happy to further discuss with you.
>
> Best,
>
> Authors

---

> ### Author Response · Authors · 2025-11-26
> **A kind reminder of discussion : We would like to learn Reviewer’s opinion and address any remaining concerns.**
>
> Dear Reviewer nHoy,
>
> We would kindly like to inquire if you would get a chance to review our response and if there are any remaining questions we can address.
>
> Your insights, both the constructive suggestions and areas of contention, have been crucial for us. We have done our best to address each point and clarify any misunderstandings. We are truly keen to have a constructive dialogue with you to refine our work further.
>
> Best regards,
>
> Authors

---

> > ### Comment · Reviewer_nHoy · 2025-11-26
> >
> > Thank you for your detailed rebuttal. However, two fundamental technical issues remain unresolved, which are crucial for assessing the theoretical validity of the method: the mathematical validity of the "projection equivalent vector $\alpha$", and the necessity and significance of the regularization term $\lambda‖\alpha−1‖^2$.
> >
> > ---
> >
> > ## **1. The mathematical basis for considering $\alpha$ as a "projection equivalent vector" remains unclear.**
> >
> > The paper’s central theoretical claim is: $(W - W_q)X \in \operatorname{Null}(X)$
> >
> > and that projecting the quantization error into the null space alleviates output error.
> >
> > However, a true projection operator ($P_{\text{null}}$) must satisfy:
> >
> > * idempotency: $(P^2 = P)$
> > * (if orthogonal) symmetry: $(P = P^\top)$
> >
> > Yet in the implementation, the projection matrix is replaced by: $P_{\text{null}} \approx \alpha \in \mathbb{R}^{C_{\text{out}}}$, i.e., a per-output-channel scaling vector.
> >
> > From a strict linear algebra perspective, **a per-channel scaling cannot represent or approximate any projection operator**, nor can it ensure that $(W - W_q)X$ lies in the null space for general $X$.
> >
> > In the rebuttal, the authors justify this substitution based on deployment constraints and memory overhead.
> > However, this does **not** address the core conceptual question:
> >
> > **Does $\alpha$ retain any property of the projection operator at all?**
> >
> > If not, what mechanism is it actually implementing?
> >
> > If $\alpha$ is fundamentally not a projection operator (nor a linear approximation of one), the theoretical motivation and the actual algorithm are misaligned, and this misalignment must be explicitly clarified.
> >
> > ---
> >
> > ## **2. The motivation and effect of the regularization term $\lambda|\alpha−1|^2$ are still not theoretically justified**
> >
> > The closed-form solution introduces a regularization term: $\lambda|\alpha - \mathbf{1}|^2.$
> >
> > In the rebuttal, this is justified as “preventing interference with existing PTQ corrections and ensuring stability.”
> > However, this explanation is incomplete for several reasons:
> >
> > - **If $\alpha$ represents a projection-equivalent vector, why should it be constrained to remain close to the identity vector?**
> >
> >     Projection operators do not inherently resemble the identity.
> > Constraining α to be close to 1 strongly suggests that the method is performing **scale correction** rather than any form of projection.
> >
> > - **Without ablations for $\lambda=0$, the necessity of the regularizer remains unproven.**
> >
> >     Critical questions remain unanswered:
> >
> >     * Does $\lambda=0$ cause overfitting to the calibration dataset?
> >     * Does $\alpha$ become numerically unstable without regularization?
> >     * Does $\lambda$ effectively collapse $\alpha$’s expressiveness, making it closer to minor scale tuning?
> >     * How does λ interact with GPTQ/QuIP scaling or rounding corrections?
> >
> > Without evidence, the role of this regularizer—and therefore the nature of α itself—remains ambiguous.

---

> > > ### Author Response · Authors · 2025-11-27
> > > **Response to Reviewer nHoy R2**
> > >
> > > Dear Reviewer nHoy,
> > >
> > > Thank you very much for your on-time discussion. We will provide detailed answers for addressing your remaining issues.
> > >
> > > ---
> > > > **RQ1.** $\alpha$ is not a projection operator and does not retain projection properties.
> > >
> > > **A1.** We completely agree with the your observation. Our $\alpha$ is indeed not a projection operator, because our goal is never to approximate the null space projection matrix $\Delta$ itself with a projection vector. We must clarify that, our objective function is
> > >
> > > **${argmin}_{\alpha}\|(W-W_q)\times\Delta - (W-\alpha W_q)\|^2$**.
> > >
> > > This means that **by deriving a scale vector $\alpha$ and applying it on $W_q$, the quantization error introduced by $(W-\alpha W_q)$ behaves as closely as possible to the original quantization error optimized by the null space projection, i.e., $(W-W_q)\times\Delta$**. Because after deployment there will be only the quantized weights $W_q$ on hardwares, computing a projection matrix to apply on $W_q$ will convert the quantized model back to FP16, as we describe in Line 223-226. Therefore, the operation $(W - W_q)\times\Delta$ is **impractical for deployment and cannot be executed**. Thus, we **reformulate** the problem into let the final deployed weights $\alpha W_q$ to approach the best possible weights whose quantization error fully lied into the null space of input activations, i,e., $\alpha W_q \approx W-(W - W_q)\times\Delta$. Therefore, **$\alpha$ is not a projection operator, and does not approximate $\Delta$, but the optimal linear surrogate that transfers the effect of null space projection onto the deployable quantized weights $W_q$.** We will further elaborate on this point in the revised manuscript to avoid confusion.
> > >
> > > ---
> > > > **RQ2.** The motivation and effect of $\lambda$.
> > >
> > > **A2.** Thanks for your thoughtful comment. As clarified above, $\alpha$ is not a projection operator, so the regularizer is not intended to preserve the projection property. Because the baselines optimized by the traditional paradigm that minimize the numerical quantization error $\||WX-W_qX\||^2$ have already converged and rely on channelwise or groupwise scaling, If no regularization is applied, $\alpha $ may introduce large scaling deviations that disrupt these existing PTQ optimizations and produce undesirable shifts. Thus, $\lambda$ is introduced to **ensure null space optimization while preserving the effectiveness of the original constraint. In other words, the regularizer balances the two optimization goals simultaneously.** To verify the necessity of $\lambda$, we conduct ablations as the table below. We find that when $\lambda=0$, some performance even degrade below the un-optimized baseline, indicating that $\alpha$ without $\lambda$ damages the original PTQ corrections. Even get improvements in some cases, the gains are consistently smaller than those with regularization. These observations also address your remaining questions: *1)* $\lambda=0$ will not cause calibration overfitting, because our calibration set selection c4 or wikitext matches the pretraining data which may cover the generalization ability, but will lead to unstable scaling. *2)* $\lambda=0$ will increase variance and cause erratic scaling. *3)* The regularizer does not restrict $\alpha$ from performing meaningful corrections but only discourages extreme or disruptive behaviors. *4)* $\lambda=0$ will inevitably conflicts with the original PTQ correction mechanisms.
> > >
> > > |Model|q2n|wiki|ptb|c4|
> > > |:-|:-|:-|:-|:-|
> > > |Llama3.3-70b|-|18.34|25.04|21.67|
> > > ||$\alpha$ wo $\lambda$|17.75|24.47|21.05|
> > > ||$\alpha$ w $\lambda$|**17.58**|**24.90**|**20.95**|
> > > |Llama3.1-70b|-|18.97|27.61|21.97|
> > > ||$\alpha$ wo $\lambda$|19.07|26.89|22.09|
> > > ||$\alpha$ w $\lambda$|**17.42**|**24.30**|**20.64**|
> > > |Llama3-70b|-|22.39|30.17|26.89|
> > > ||$\alpha$ wo $\lambda$|21.83|29.32|25.09|
> > > ||$\alpha$ w $\lambda$|**19.92**|**28.80**|**23.72**|
> > > |Llama3.1-8b|-|24.55|36.72|26.79|
> > > ||$\alpha$ wo $\lambda$|22.20|31.47|23.26|
> > > ||$\alpha$ w $\lambda$|**17.79**|**25.77**|**20.01**|
> > > |Llama3-8b|-|19.83|33.96|23.08|
> > > ||$\alpha$ wo $\lambda$|24.29|29.38|23.47|
> > > ||$\alpha$ w $\lambda$|**17.55**|**26.67**|**20.20**|
> > > |Qwen2.5-32B|-|23.23|26.33|18.29|
> > > ||$\alpha$ wo $\lambda$|17.85|25.21|17.50|
> > > ||$\alpha$ w $\lambda$|**15.83**|**23.51**|**17.83**|
> > > |Qwen3-32B|-|37.10|54.97|26.59|
> > > ||$\alpha$ wo $\lambda$|34.34|49.82|25.92|
> > > ||$\alpha$ w $\lambda$|**26.98**|**42.97**|**22.72**|
> > >
> > > ---
> > > Finally, we would like to re-emphasize that the equivalent vector $\alpha$ is **not** a projection operator, and we do not use it to approximate $\Delta$ to project the quantization error. Instead, by applying channel-wise scaling to deployable $W_q$, we make the quantization error introduced by $\alpha W_q$ closely matches the effect of applying the projection matrix $\Delta$ to the original perturbation $(W-W_q)$.
> > >
> > > ---
> > > We have done our best to address any misunderstandings and we are very happy to provide any additional clarifications that you may need.
> > >
> > > Best,
> > >
> > > Authors

---

### Official Review · Reviewer_bpHa · 2025-10-31

**Soundness:** 3
**Presentation:** 3
**Contribution:** 3
**Rating:** 6
**Confidence:** 4

**Summary:**

This paper introduces a novel perspective for post-training quantization (PTQ) of LLMs by leveraging null space optimization. The core idea is to constrain the quantization error to the null space of input activations, thereby minimizing the impact on the model's output. The authors propose Q2N, a practical method that includes an efficient approximation of the null space and a closed-form "equivalent projection vector" `α` that can be absorbed into existing scaling factors to avoid any inference overhead. Experiments show that Q2N consistently improves performance when added to state-of-the-art PTQ baselines like GPTQ and QuIP.
The paper is well-written, theoretically sound, and makes an original contribution. The core concept of using the null space is a valuable perspective that opens a new, principled direction for PTQ research. The experimental methodology is robust, providing evidence for the method's effectiveness across a range of models and baselines.

**Strengths:**

(1) Novel Conceptual Framework: The idea of guiding quantization error into the null space is original and reframes the PTQ problem in a more targeted way than simply minimizing numerical error.
(2) Practical and Efficient Algorithm: The paper translates theory into a practical algorithm, with the memory-free `α` vector being a key element that makes the method viable for deployment without inference overhead.
(3) Comprehensive Validation: The approach is thoroughly evaluated on multiple modern LLMs,  demonstrating its general applicability and robustness.

**Weaknesses:**

(1) Marginal Gains vs. Complexity: While gains are consistent, they can be modest. A more direct analysis of the trade-off between the one-time quantization complexity and the magnitude of performance improvement would be beneficial.
(2) Hyperparameter Sensitivity: The paper would be strengthened by a more detailed sensitivity analysis for the key hyperparameters `t` and `λ` to better guide practitioners on their selection.
(3) Limited Bit-Width Scope: While the paper focuses on 2-3 bit quantization where the method shows clear benefits, it would be valuable to understand the method's behavior across a wider range of bit-widths (e.g., 4-6 bits) to better characterize when null space optimization provides the most value.
(4) Indirect Evaluation of Null Space Quality: While Table 3 and Table 4 compare different null space estimation methods through downstream performance, the paper would benefit from direct mathematical metrics (e.g., subspace distance, projection error, or spectral analysis) to quantitatively assess how well the Prefix-Suffix Sum Ratio approximation captures the true null space.

**Questions:**

(1) The paper focuses on low-bit quantization. Have you explored the effectiveness of Q2N at higher bit-widths, such as 4-bit and above? Does the method continue to provide consistent improvements in these less aggressive quantization regimes?
(2) Could you provide a sensitivity analysis plot for hyperparameters `t` and `λ` against a key performance metric (e.g., perplexity)?
(3) The paper uses groupsize=128 for per-group quantization. How does the equivalent projection vector α interact with group-wise scaling factors? Have you explored the sensitivity to different groupsizes, and does the method's effectiveness vary with finer or coarser granularity?

---

> ### Author Response · Authors · 2025-11-22
> **Response to Reviewer bpHa**
>
> Dear Reviewer bpHa,
>
> Thanks for your positive and insightful reviews which help us strengthen the manuscript. We have provided answers below with the hope of offering a better understanding.
>
> ---
> >**W1.** Tradeoff between quantization complexity and performance improvement.
>
> **A1.** Using per-group quantization with progressively smaller group sizes increases one time quantization complexity. We examine how our method’s performance changes under this increased complexity on 2bit Qwen3-32B, as the table shown.We observe that when groupsize = -1 (per-channel), the perplexity almost collapses, yet the improvement is also the largest. When groupsize=256, we still get a 50% improvement. This confirms that the worse the initial performance is, the more improvements our method gets. Nevertheless, we recommend to apply our method at groupsize 128, where it achieves strong performance without introducing noticeable additional overhead.
> |Q2N|G|arc c|arc e|hella|mmlu|piqa|wino|wiki|c4|
> |:-|:-|:-|:-|:-|:-|:-|:-|:-|:-|
> |✘|-1|24.4|24.76|24.19|24.08|49.95|48.49|9.3e3|3.2e3|
> |✔||**26.79**|**25.25**|**24.64**|**24.56**|**51.52**|**49.33**|**2.8e3**|**2.3e3**|
> |✘|256|27.13|29.5|37.71|24.53|54.41|50.75|82.83|48.25|
> |✔||**27.65**|**29.42**|**43.94**|**24.18**|**56.47**|**51.38**|**41.70**|**30.81**|
> |✘|128|25.43|32.45|47.85|24.88|57.5|51.46|37.1|26.59|
> |✔||**28.5**|**35.73**|**53.09**|**25.24**|**59.85**|**53.59**|**26.98**|**22.72**|
>
> ---
> >**W2.** Hyperparameter sensitivity
>
> **A2.** Thank you for your suggestion. We conduct extensive experiments on Llama, Qwen, and DeepSeek using multiple baselines to explore the optimal hyperparameter range. The search range for λ is 0.1 to 0.9, and t ranges from 0.05 to 0.2. **We visualize the results using a heatmap to identify suitable hyperparameter regions, please refer to Figure 5 in Appendix C in our new version PDF**. We observe that the optimal choice of λ falls within the range of 0.1 to 0.4 and t is 0.05 to 1, which means that our method is robust in such range. We attribute the performance degradation at higher threshold t to the fact that increasing the threshold causes the split point obtained by our PSSR method to shift significantly toward the long tail of the eigenvalue distribution, resulting in poor estimation of the effective eigenvalues. Similarly, the performance decrease at larger λ occurs because a higher λ forces the elements of the equivalent projection vector to approach 1, which makes the projected weight matrix remain unchanged.
>
> ---
> >**W3.** Limited bitwdith scope
>
> **A3.** Our method primarily focuses on low-bit scenario to show the effectiveness of null space optimization, because advanced PTQ techniques such as GPTQ have already achieved **near-lossless performance at 4-6 bits**, leaving limited value for further exploration in this range. So in our view it is unnecessary to further improve high-bit performance. Nevertheless, our method still provides a slight improvement at 4 bits on both Llama3 and Qwen3.
>
> |Model|bit|Q2N|wiki|ptb|c4|
> |:-|:-|:-|:-|:-|:-|
> |llama3-8b|4|✘|7.90|12.47|10.43|
> |||✔|**7.53**|**12.12**|**10.20**|
> ||6|✘|6.20|10.64|**8.94**|
> |||✔|6.20|10.64|8.95|
> |qwen3-32b|4|✘|8.38|13.88|11.37|
> |||✔|**8.31**|**13.27**|**11.36**|
> ||6|✘|7.63|**12.71**|10.81|
> |||✔|7.63|12.73|**10.80**|
>
> ---
> >**W4.** Indirect Evaluation of Null Space Quality
>
> **A4.** Thank you for your considerable comment. Since the activations of LLMs rarely exit zero eigenvalues, we **cannot obtain the true null space and can only get an approximated one**. To assess the accuracy of our approximation using mathematical metric, we measure the Euclidean distance between the layer outputs before and after projection with the corresponding full-precision outputs. A smaller distance indicates that part of the quantization error has been effectively projected into the null space. As shown in the table where we select the 1st layer of llama3-8b, our method achieves smaller distance with FP outputs.
> |Q2N|q|k|v|o|gate|up|down|
> |:-|:-|:-|:-|:-|:-|:-|:-|
> |✘|14.94|10.69|1.87|0.41|12.00|7.47|13.07|
> |✔|**8.89**|**6.27**|**1.05**|**0.13**|**7.89**|**3.65**|**8.14**|
>
> ---
> >**Q1.** High-bit evaluation
>
> **AQ1.** Please refer to A3 for detailed.
>
> ---
> >**Q2.** Sensitivity analysis of hyperparameters
>
> **AQ2.** Please refer to A2 for detailed.
>
> ---
> >**Q3.** How does the projection vector interact with groupwise scaling factor. Explore the sensitivity of different groupsize.
>
> **AQ3.** The equivalent projection vector is element-wise multiplied with the matrix of per-group scaling factors, which means that each per-group scaling factor within a channel is multiplied by the same corresponding element of the projection vector. For the sensitivity of different groupsize, please refer to A1 for detailed.
>
> ---
> Thank you again for your reviews! It will be encouraging that you can raise your score if we have addressed all your issues. And we are also very happy to further discuss with you.
>
> Best,
>
> Authors

---

> ### Author Response · Authors · 2025-11-26
> **A kind reminder of discussion : We would like to learn Reviewer’s opinion and address any remaining concerns.**
>
> Dear Reviewer bpHa,
>
> We would kindly like to inquire if you would get a chance to review our response and if there are any remaining questions we can address.
>
> Your insights, both the constructive suggestions and areas of contention, have been crucial for us. We have done our best to address each point and clarify any misunderstandings. We are truly keen to have a constructive dialogue with you to refine our work further.
>
> Best regards,
>
> Authors

---

> > ### Comment · Reviewer_bpHa · 2025-11-28
> > **Response to Author**
> >
> > We appreciate the authors' comprehensive responses to our comments. After careful deliberation, considering the work's novelty, methodological rigor, and potential impact on the field, we have decided to maintain our original score. Thanks!

---

### Meta-Review · Area_Chair_Sxs4 · 2026-01-06

**Summary:**

This paper focuses on the post-training quantization, and introduces the concept of null space into LLMs quantization. The main argument is that the quantization error could be alleviated effectively by the null space constraints. The proposed method is validated on several milestone post-training quantization baselines, which demonstrate a further improvement boost. The reviewers acknowledged the novel conceptual framework, the compatibility of the null-space-based method with existing quantized inference pipelines, as well as the experimental verification.

However, it seems some reviewer concerns still remain after the discussion phase and are explicitly stated in the reviewer's response. The Reviewer nHoy is concerned about the mathematical basis for considering $\alpha$ as a "projection equivalent vector" with the motivation of the regularization term; and Reviewer q2vZ is concerned about the practical contribution and performance given no theoretical explanation. Although the theoretical results are not necessary at all times, the author's response to the remaining issues does not fully address the reviewer's concern after a thorough reading of both the submission and the discussion. Given the scenario, it is encouraged that the authors take all the comments into consideration, and the advantages of the null-space-based method can be explained in a more principled way.

**Reviewer Concerns:**

The Reviewer nHoy is concerned about the mathematical basis for considering $\alpha$ as a "projection equivalent vector" with the motivation of the regularization term; and Reviewer q2vZ is concerned about the practical contribution and performance given no theoretical explanation (e.g., for generalization), and stated the paper does not deliver amazing outcomes as a major hesitation for raising the score.

**Reviewer Scores:**

In the discussion phase, the two reviewers who gave lower scores (4) engaged with the authors to discuss the remaining concerns on the submission. The reviewers both stated that the initial rebuttal from the authors doesn't fully address their concerns about unclear theoretical motivation and the generalization analysis. The authors do a great job of providing further discussion and evidence to support their explanation, while not directly answering the reviewer's question about the theoretical or more principled explanation. The reviewers would be hesitant to increase their score regarding the latter responses.

---

### Decision · Program_Chairs · 2026-01-26

Reject